# Universal Approximation Theorem of Networks Activated by Normalization

## Abstract

Universal approximation theorem (UAT) is the fundamental theory for deep neural networks (DNNs), showing the powerful representation capacity of DNNs in approximating any function. The analyses and proofs of UAT are based on a traditional network with only linear and nonlinear activation layers, but omitting normalization layers which are commonly used for benefiting the training of modern networks. This paper conducts research on UAT of DNNs with normalization layers for the first time. We theoretically prove an infinitely wide network—with parallel layer normalizations (PLN) and linear layers only—has universal approximation capacity. We further investigate the minimum neurons required for approximate $L$-Lipchitz continuous functions, with a single hidden-layer network. We compare the approximation capacity of PLN with traditional activation functions, both in theory and by experiments. We also show PLN's approximation capacity in CNN and Transformer by experiments.

## 1. Introduction

Deep neural networks (DNNs) are widely used and have achieved excellent performance in various fields. One key theorem is that DNN is proved to have universal approximation capabilities. Cybenko (1989) proved a single hidden-layer neural network with infinite widths using sigmoidal functions has universal approximation ability. It was then extented to arbitrary bounded and nonconstant activation function (Hornik, 1991). Based on the work about the density of superpositions of a sigmoidal function in $[0, 1]^n$ (Cybenko, 1989), Barron (1993) analyzed the approximation bound of these superpositions. It was then extended to the cases of arbitrary depth (Gripenberg, 2003), bounded depth and bounded width (Maiorov & Pinkus, 1999), and the question of minimal possible width (Park et al., 2020). Besides,

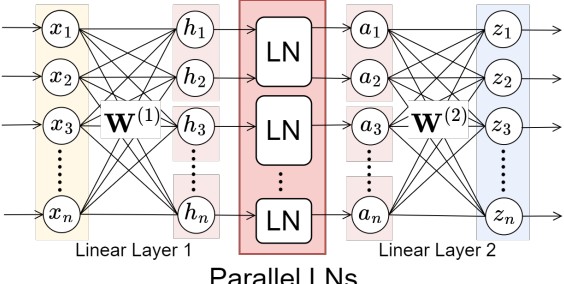

Figure 1. Illustration of a network with linear layer and Parallel layer normalizations (PLN). PLN is used on hidden neurons and divides the neurons into different partitions and conducts LN within each partition.

there were previous work studying the expressive power of neural networks form the perspective of linear regions (Montufar et al., 2014) and VC dimension (Bartlett et al., 2019).

While a DNN is able to perform excellently with its powerful representation capacity in theory, it is hard to train a DNN in practice. Normalization (Ioffe & Szegedy, 2015; Ba et al., 2016) is a ubiquitous technique in DNN, proposed for enabling varies neural networks to train effectively. The main theoretical arguments for normalization are that it can stabilize the training by its scale-invariant property (Ba et al., 2016; Arora et al., 2019; Huang et al., 2023) and accelerate the training by improve the conditioning of the optimization problem (Cai et al., 2019; Santurkar et al., 2018; Karakida et al., 2019; Ghorbani et al., 2019; Daneshmand et al., 2020; Lyu et al., 2022). However, theoretically analyzing the complexity measure (e.g., VC dimensions or the number of linear regions) of the representation capacity of neural networks with normalization is a challenging task, because normalized networks do not follow the assumptions for calculating linear regions or VC dimensions (Huang et al., 2021).

As a recent work, Ni et al. (2024) revealed that layer normalization (LN) contains nonlinearity itself. They constructed a network with layerwise composition of linear and LN transformations, referred to as LN-Net. They theoretically show that, given $m$ samples with any label assignment, an LN-Net with only 3 neurons in each layer and $O(m)$ LN layers can correctly classify them. Furthermore, they figured out that given an LN-Net $f_\theta(\cdot)$ with width 3 and depth $L$, its

[1]Anonymous Institution, Anonymous City, Anonymous Region, Anonymous Country. Correspondence to: Anonymous Author <anon.email@domain.com>.

Preliminary work. Under review by the International Conference on Machine Learning (ICML). Do not distribute.

VC dimension $VCdim(f_\theta(\cdot))$ is lower bounded by $L + 2$. All the work above revealed one interesting conjecture—normalization is possible for representation directly, rather than for optimization only in the previous DNNs.

Inspired by the work in (Ni et al., 2024), we shift our perspective from deep networks for classification, to wide networks for approximation. We focus on parallel layer normalizations (PLN) rather than serial LN-Net, as shown in Figure 1. We theoretically prove an infinitely wide network—with a "linear-PLN-linear" structure—has universal approximation ability on $[0,1]^n$. This theorem has given us new inspirations: can we take normalization as activation layers in DNNs? When we discuss about activation layers, is there something interesting about optimization?

Considering the width-bounded networks, one interesting question is that: can normalization reach the comparable expressive capacity of the traditional activations with limited neurons? The answer is yes. We consider approximating any $L$-Lipchitz function on $[0,1]$ by the $L^\infty$ error $\varepsilon$, with a single hidden-layer network. We mathematically find the minimum of the required neurons using PLN is no more than $d(\lfloor L/2\varepsilon \rfloor + 1)$, where $d$ is the size of each LN in PLN. This width can decrease to only twice that of using ReLU. The results above are obtained in theory, it is not the same in practical training, for the optimization process is also of great importance. We also conduct approximation experiments to identify this multiple relationship. Beyond our prediction, we find that PLN performs better than ReLU in approximation. We conclude that taking PLN as an activation layer is feasible completely.

We also conduct experiments to apply PLN in CNN and Transformer architectures. To begin with, we verify that PLN can replace the combination of activation functions and normalizations in DNNs. PLN can perform well with only linear layers, for it has ability of both representation and optimization. Then we take PLN as Normalization and explore the performances of different activation functions. We find that activation functions may not be necessary in classification task when using CNN with PLN as Normalization. As for machine translation task using Transformer with PLN as Normalization, activation functions remain important, for translation may require stronger nonlinear representation capacity. Besides, we find that the combination of PLN and ReLU performs exceptionally well in our experiment settings.

## 2. Preliminary and Notation

We use a lowercase letter $x \in \mathbb{R}$ to denote a scalar, boldface lowercase letter $\mathbf{x} \in \mathbb{R}^n$ for a vector and boldface uppercase letter for a matrix $\boldsymbol{X} \in \mathbb{R}^{d \times n}$, where $\mathbb{R}$ is the set of real-valued numbers, and $d, n$ are positive integers. Following (Cybenko, 1989), the definition of a sigmoidal function is

shown as below.

**Definition 1** (Sigmoidal function). *$\sigma$ is a sigmoidal function, if $\sigma(-\infty) = 0$, and $\sigma(+\infty) = 1$.*

Here we show one version of universal approximation theorem—Theorem 4 in (Cybenko, 1989) as follows.

**Theorem 1** (Universal Approximation Theorem). *Let $\sigma$ be bounded measurable sigmoidal function. The finite sums of the form*

$$G(\mathbf{x}) = \sum_{j=1}^{N} \alpha_j \sigma(\boldsymbol{w}_j^\top \mathbf{x} + b_j) \qquad (1)$$

*are dence in $C([0,1]^n)$. In other words, given any $f \in C([0,1]^n)$ and $\varepsilon > 0$, there is a sum, $G(\mathbf{x})$, of the above form, for which*

$$|G(\mathbf{x}) - f(\mathbf{x})| < \varepsilon, \forall \mathbf{x} \in [0,1]^n. \qquad (2)$$

**Layer Normalization.** Layer Normalization (LN) is an essential layer in modern deep neural networks mainly for stabilizing training. Given a single sample of layer input $\mathbf{x} = [x_1, x_2, \cdots, x_d] \in \mathbb{R}^d$ with $d$ neurons in a neural network, LN standardizes $\mathbf{x}$ within the neurons as [1]:

$$\hat{x}_j = LN(x_j) = \frac{x_j - \mu}{\sigma}, \;\; j = 1, 2, \cdots, d, \qquad (3)$$

where $\mu = \frac{1}{d} \sum_{i=1}^{d} x_j$, $\sigma = \sqrt{\frac{1}{d} \sum_{i=1}^{d} (x_j - \mu)^2}$ are the mean and variance for each sample respectively.

**Parallel Layer Normalizations.** Given $\mathbf{x}_1 \in \mathbb{R}^{d_1}, \mathbf{x}_2 \in \mathbb{R}^{d_2}, \cdots, \mathbf{x}_N \in \mathbb{R}^{d_N}$, and each $d_i \geq 2$. For the input $[\mathbf{x}_1^\top, \cdots, \mathbf{x}_N^\top]^\top$, we define a calculation as parallel layer normalizations (PLN), if the output $[\hat{\mathbf{x}}_1^\top, \cdots, \hat{\mathbf{x}}_N^\top]^\top$ satisfies $\hat{\mathbf{x}}_i = LN(\mathbf{x}_i)$ for $1 \leq i \leq N$. Specially, if $d_1 = d_2 = \cdots = d_N = d$, we refer such PLN as PLN-$d$. We say $d$ is the norm size of PLN-$d$.

## 3. Normalization for Universal Approximation

In this section, we will first show how to approximate any continuous function on $[0,1]^n$ by taking LN as an activation layer. We then extend the result to a neural network with PLN and linear layers only. Finally, we further disocuss the approximation on LN without centering, namely Layer Scaling (LS) or RMSNorm (Zhang & Sennrich, 2019).

### 3.1. LN for Universal Approximation Theorem

**Definition 2** (Representable function class). *Given $\varphi : \mathbb{R}^d \to \mathbb{R}^d$, we define $\mathcal{G}(N; \varphi)$ as a representable function*

---

[1]LN usually uses extra learnable scale and shift parameters (Ioffe & Szegedy, 2015), and we omit them for simplifying discussion as they are affine transformation in native

*class—we say $G(\mathbf{x}) \in \mathcal{G}(N; \varphi)$ where $\mathbf{x} \in \mathbb{R}^n$, if there are some $\boldsymbol{\alpha}_j, \boldsymbol{b}_j \in \mathbb{R}^d, \boldsymbol{W}_j \in \mathbb{R}^{d \times n}$ for each $j$, such that*

$$G(\mathbf{x}) = \sum_{j=1}^{N} \boldsymbol{\alpha}_j^\top \varphi(\boldsymbol{W}_j \mathbf{x} + \boldsymbol{b}_j). \tag{4}$$

Here we show how to apply LN to approximate any continuous function on $[0, 1]^n$.

**Theorem 2** (LN for Universal Approximation Theorem). *Let $LN(\cdot)$ be Layer Normalization on $\mathbb{R}^d, d \geq 2$. Given any $f \in C([0, 1]^n)$ and $\varepsilon > 0$, there is a sum $G(\mathbf{x}) \in \mathcal{G}(N; LN)$ when $N$ is large enough, subjected to $|G(\mathbf{x}) - f(\mathbf{x})| < \varepsilon$ for $\mathbf{x} \in [0, 1]^n$.*

To prove the theorem, we first give Lemma 1 as follows.

**Lemma 1.** *There is a $G(\mathbf{x}) \in \mathcal{G}(N + 1; LN)$, subjected to that $G(\mathbf{x})$ is a linear combination with $N$ bounded measurable sigmoidal functions.*

*Proof.* Here we give the proof at the case $d = 2$.

Assume that $G(\mathbf{x}) = \sum_{j=1}^{N+1} \boldsymbol{\alpha}_j^\top LN(\boldsymbol{W}_j \mathbf{x} + \boldsymbol{b}_j)$. Let $\boldsymbol{\alpha}_j = [\hat{\alpha}_j, 0]^\top, \boldsymbol{W}_j = [\boldsymbol{w}_j, -\boldsymbol{w}_j]^\top$ and $\boldsymbol{b}_j = [b_j, -b_j]^\top$ for $1 \leq j \leq N$. Besides, let $\boldsymbol{\alpha}_{N+1} = [(\hat{\alpha}_1 + \cdots + \hat{\alpha}_N), 0]^\top, \boldsymbol{W}_{N+1} = \boldsymbol{O}$ and $\boldsymbol{b}_{N+1} = [1, -1]^\top$. Then by Eqn.3, it is easy to identify that $\boldsymbol{W}_j \mathbf{x} + \boldsymbol{b}_j = [\boldsymbol{w}_j^\top \mathbf{x} + b_j, -(\boldsymbol{w}_j^\top \mathbf{x} + b_j)]^\top$ for $1 \leq j \leq N$, while $[1, -1]^\top$ for $j = N + 1$. Here we have

$$G(\mathbf{x}) = \sum_{j=1}^{N} \hat{\alpha}_j \cdot \frac{\boldsymbol{w}_j^\top \mathbf{x} + b_j}{\sqrt{(\boldsymbol{w}_j^\top \mathbf{x} + b_j)^2}} + \sum_{j=1}^{N} \hat{\alpha}_j$$

$$= \sum_{j=1}^{N} 2\hat{\alpha}_j \left[ \frac{\boldsymbol{w}_j^\top \mathbf{x} + b_j}{2|\boldsymbol{w}_j^\top \mathbf{x} + b_j|} + \frac{1}{2} \right] \tag{5}$$

$$= \sum_{j=1}^{N} 2\hat{\alpha}_j \sigma(\boldsymbol{w}_j^\top \mathbf{x} + b_j),$$

where $\sigma(x) = (x/|x| + 1)/2$ is easy to identify as a bounded measurable sigmoidal function. $\square$

Lemma 1 also holds for the case $d > 2$, please refer to *Appendix A.1* for more details. By Lemma 1 and Theorem 1, we can prove Theorem 2.

In this subsection, we have shown how to approximate any continuous function on $[0, 1]^n$ by taking LN as an activation function. In the next subsection, we will provide details on how to apply PLN in a neural network.

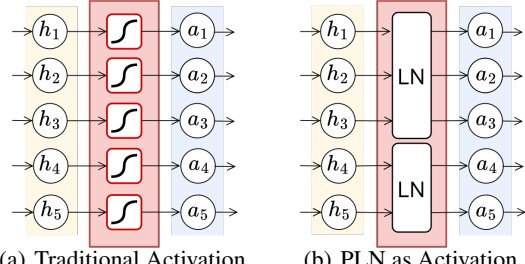

(a) Traditional Activation.     (b) PLN as Activation.

*Figure 2.* Tradition activations act on each neuron, while PLN requires a group of neurons to activate, where. Besides, the norm sizes in PLN can be different.

### 3.2. Parallel LNs in Networks

Theorem 1 describes a neural network with single hidden-layer, so does ours. Different from the traditional activation functions, PLN activate each group of neurons, rather than neuron. The intuitive difference is shown in Figure 2.

**General Activation Functions.** As shown in Figure 2, traditional activation functions act on single neuron. Based on Theorem 2, we believe that a more general activation function can be defined on more neurons. There will be some meaningful interactions within these neurons. In fact, there is already such an activation function—softmax—which is widely used in attention layers (Vaswani et al., 2017). Softmax is first used for multi-class classification tasks, coming from binary classification tasks with sigmoid. Layer Normalization (Ba et al., 2016) is also such an activation function, but its nonlinearity is clearly figured out eight years after its proposition (Ni et al., 2024). We think such general activation functions are also of great importance in a neural network.

**Connection with LN-G.** Specially, when the norm size of each LN equals to $d$, namely when we get PLN-$d$, it has the same structure as LN-G (Ni et al., 2024)—which divides neurons of a layer into groups and performs LN in each group in parallel. LN-G focuses on grouping from a wide LN, while PLN focuses on filling narrow LNs to reach the network width. Although PLN-$d$ has the same structure as LN-G, its concept leans more towards activation functions. Like ReLU, we are not concerned about the width of a network, but treat each neuron or each $d$ neurons as an activation unit.

Based on the discussion above, we find that PLN comes from normalizations, but behaves like an activation function more. We then extend Theorem 2 to neural networks with PLN and linear layers only.

**Corollary 1** (Universal Approximation Theorem of Neural Networks Activated by PLN). *Any continuous function on $[0, 1]^n$ can be approximated at any precision, by an infinitely wide network with only linear layers and PLN.*

However, PLN indeed requires more neurons for once activation than the traditional activation functions. Can we activate the neurons more efficiently than LN? Scaling only is a feasible choice to replace LN, as discussed in the next subsection.

### 3.3. RMSNorm for Universal Approximation Theorem

Ni et al. (2024) show the nonlinearity of LN exists only in scaling. When focusing exclusively on representation capacity, centering is not a necessary part when PLN serves as the activation function. Therefore, scaling only—namely RMSNorm (Zhang & Sennrich, 2019)—may suffice for universal approximation.

Here, we remove the centering in LN to obtain Layer Scaling (LS)—LS standardizes $\mathbf{x}$ across the neurons as:

$$\hat{x}_j = LS(x_j) = \frac{x_j}{\sqrt{\overline{x^2}}}, \quad j = 1, 2, \cdots, d, \qquad (6)$$

where $\overline{x^2} = \frac{1}{d} \sum_{i=1}^{d} x_j^2$ is the second-order moment for each sample, rather than the variance.

Similarly, we can also construct parallel LSs (PLS) for the universal approximation theorem.

**Corollary 2** (LS for Universal Approximation Theorem). *Let $LS(\cdot)$ be Layer Scaling (or RMSNorm) on $\mathbb{R}^d, d \geq 1$. Given any $f \in C([0,1]^n)$ and $\varepsilon > 0$, there is a sum $G(\mathbf{x}) \in \mathcal{G}(N; LS)$ when $N$ is large enough, subjected to $|G(\mathbf{x}) - f(\mathbf{x})| < \varepsilon$ for $\mathbf{x} \in [0,1]^n$.*

The proof is similar to that of Theorem 2, please refer to *Appendix* A.2 for details. By Corollary 2, we point out that centering is not necessary for approximation. Therefore, the extreme case of PLS is PLS-1, which activates each neuron similarly to traditional activation functions.

**Conclusion**   In this section, we have proved the universal approximation theorem for an infinitely wide neural network with one hidden-layer, whose activation function is based on normalizations (PLN or PLS). One practical question is: What is the representation capacity of bounded-width networks? This will be discussed in the following section, where we will also compare it with other traditional activation functions.

## 4. Approximation by Bounded-wide Networks

In this section, we will compare the representation capacity of different activation functions in single hidden-layer networks. We focus on approximating $L$-Lipschitz continuous functions on $[0,1]$ rather than arbitrary functions on $\mathbb{R}^n$, for simplification and visualization. We will show the comparison results both theoretically and experimentally.

### 4.1. Approximation Bound

Given a single hidden-layer neural network, how many neurons are required for universal approximation with different activation functions? We will answer this question in this subsection, including sigmoid, tanh, ReLU, PLN, and PLS.

**Definition 3** (Approximation Bound). *We denote $\mathcal{F}(I; L)$ as a set consisting of all the $L$-Lipschitz continuous functions $f \in C(I)$. Given $\mathcal{G}(N; \varphi)$, where $\varphi : \mathbb{R}^d \to \mathbb{R}^d$. Here we define*

$$\mathcal{N}(\varphi) = \inf_N \left\{ N : \sup_{f \in \mathcal{F}} \inf_{g \in \mathcal{G}} \|f - g\| < \varepsilon \right\}, \qquad (7)$$

*as the minimum $N$ to approximate $\mathcal{F}$ by $\mathcal{G}$ on $I$ with error bound $\varepsilon$. Here $\|f - g\| = \sup_{\mathbf{x} \in I} |f(\mathbf{x}) - g(\mathbf{x})|$.*

Besides, we define $d_{\min}(\varphi)$ as the minimum $d$, subjected to that $\varphi$ can be defined on $\mathbb{R}^d$. For example, we have $d_{\min}(ReLU) = 1$ and $d_{\min}(LN) = 2$. Then we denote $\mathcal{W}(\varphi) = d_{\min}(\varphi)\mathcal{N}(\varphi)$ as the minimum width of the corresponding network.

Without loss of generality, we set $I = [0,1]$ as default. Here we give the approximation bound of LN and LS.

**Proposition 1** (Approximation Bound of LN). *Given $\mathcal{F} = \mathcal{F}([0,1]; L)$ and $\mathcal{G} = \mathcal{G}(N; LN)$, where $LN(\cdot)$ denotes LN on $\mathbb{R}^d, d \geq 2$. Given the error bound $\varepsilon > 0$, we have*

$$\mathcal{N}(LN) \leq \lfloor L/2\varepsilon \rfloor + 1. \qquad (8)$$

*Furthermore, we have $\mathcal{W}(LN) \leq 2(\lfloor L/2\varepsilon \rfloor + 1)$.*

For $\mathcal{N}(LS)$, it has the same upper bound with $\mathcal{N}(LN)$—but $\mathcal{W}(LS) \leq \lfloor L/2\varepsilon \rfloor + 1$, for $d_{\min}(LS) = 1$. Please refer to *Appendix* A.3 for the detailed proof.

As one of the initial functions for universal approximation, we show the upper bound of sigmoid in Proposition 2.

**Proposition 2** (Approximation Bound of Sigmoid). *Given $\mathcal{F} = \mathcal{F}([0,1]; L)$ and $\mathcal{G} = \mathcal{G}(N; \sigma)$, where $\sigma(x) = 1/(1 + e^{-x})$ denotes the sigmoid function. Given the error bound $\varepsilon > 0$, we have*

$$\mathcal{N}(\sigma) \leq \lfloor L/2\varepsilon \rfloor + 1. \qquad (9)$$

*Furthermore, we have $\mathcal{W}(\sigma) \leq 2\lfloor L/2\varepsilon \rfloor + 1$.*

As for tanh, we can easily get that $\tanh(x) = 2\sigma(2x) - 1$, it has the same conclusion with sigmoid. Please refer to *Appendix* A.4 for detailed proof.

ReLU has been one of the most widely used activation in neural networks. For its simplicity, we can get both its upper bound and lower bound easily in Proposition 3.

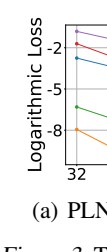
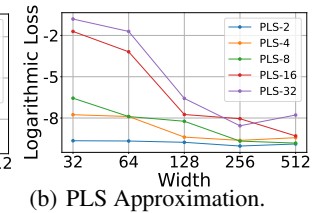

(a) PLN Approximation.    (b) PLS Approximation.

*Figure 3.* The results of logarithmic loss of PLN and PLS varying with width, using different norm sizes.

**Proposition 3** (Approximation Bound of ReLU). *Given* $\mathcal{F} = \mathcal{F}([0,1]; L)$ *and* $\mathcal{G} = \mathcal{G}(N; ReLU)$*, where* $ReLU(x) = \max(0, x)$ *denotes the ReLU function. Given the error bound* $\varepsilon > 0$*, we have*

$$\lfloor L/2\varepsilon \rfloor - 1 \leq \mathcal{N}(ReLU) \leq \lfloor L/2\varepsilon \rfloor + 2. \quad (10)$$

*Similarly, we have* $\lfloor L/2\varepsilon \rfloor - 1 \leq \mathcal{W}(ReLU) \leq \lfloor L/2\varepsilon \rfloor + 2$.

By Proposition 3, we find the bound of $\mathcal{N}(ReLU)$ is tight. It seems that ReLU can be seen as a "unit of measurement" under our approximation settings. For example, given a one hidden-layer network with fixed width, we can say the representation capacity of sigmoid is at least "one ReLU", while that of PLN-4 is at least "a quarter of ReLU".

However, in the practical training process, optimization is also an important factor for a good result. We thus conduct experiments to explore how width and norm size affect approximation in practice, in the following subsection.

### 4.2. Approximation Experiments

We conduct experiments to approximate a unary function on $[-5, 5]$ with different nonlinear layers (including sigmoid, tanh and ReLU) and PLN, PLS. We use a one-layer network with width ranging in $8, 16, \cdots, 4096$. We define the target function as $f(x) = \sin(2x+1) + \cos(x)$. For each activation function, we conduct experiments using two optimizers, Adam and SGD, with six learning rates (0.1, 0.01, ..., 1e-6), three random seeds (0, 10, 100), and four batch sizes (4, 8, 16). Among these configurations, the best experimental results were selected. Each experiment was trained for 1000 epochs.

#### 4.2.1. Norm Size Analysis

We first show the results using PLN and PLS with different widths and norm sizes, as shown in Figure 3.

By Figure 3(b), we find that PLS performs better with the smaller norm size, which is consistent with Proposition 1. While PLN is slightly different—PLN performs best at $d = 4$ rather than $d = 2$. We give two reasons as follows.

The first reason is that PLN-2 will output $\pm 1$ only by Eqn.3, which may block the gradient back propagation. While the

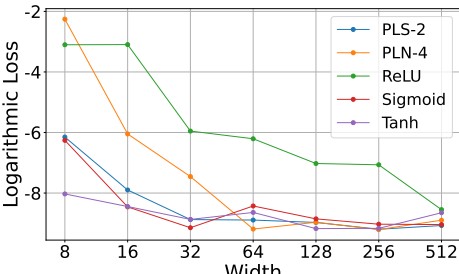

*Figure 4.* The results of logarithmic loss of different activation functions varying with width.

second linear layer will not suffer from this, ensuring that PLN-2 does not perform too badly.

The second reason is that Proposition 1 only gives the upper bound of the required neurons. As $d$ increases, we are not sure whether additional nonlinearity will be introduced.

**Trade-off between Representation and Optimization.** PLN-2 may have more representation capacity than PLN-4 in theory, but the optimization capacity is less. The same conclusion holds for PLS-1 and PLS-2 as well. Based on our analysis in subsection 4.1, PLS-1 has at least the same representation capacity as ReLU, but may suffer from gradient vanishing. We believe that there must be some trade-off between representation and optimization.

#### 4.2.2. Comparison within Different Activations

In this subsection, we will compare PLN and PLS with other activation functions by experiment, including sigmoid, tanh and ReLU. Here are the results, as shown in Figure 4.

Specifically, here we show how different activations approximate the target function intuitively in Figure 5. We find that both sigmoid and tanh perform better than ReLU, although ReLU is much more widely used at present. PLN-4 also performs well.

Actually, Maiorov & Meir (2000) denotes the lower bound of sigmoid satisfies that $\mathcal{N} \log \mathcal{N} = C/\varepsilon$, where $C$ is a constant. Combining with Proposition 3, we find that sigmoid may perform better than ReLU when approximating a Lipschitz continuous function in theory. However, as the networks get deeper, ReLU is more recommended for relieving gradient vanishing, to some extent. We can conclude that, we use ReLU in deep networks more than sigmoid or tanh for the optimization property, rather than its better expressive power. This is another finding reminding us—there may be an important correlation between representation and optimization.

We also conduct the experiments on random function. The conclusion is similar to what we obtain above. Please refer to *Appendix* B.1.2 for more details.

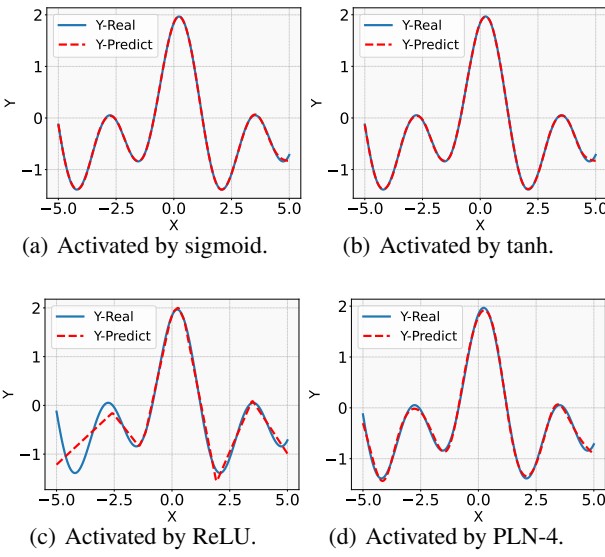

(a) Activated by sigmoid.  (b) Activated by tanh.

(c) Activated by ReLU.  (d) Activated by PLN-4.

*Figure 5.* The intuitive performance of approximating $f(x) = \sin(2x+1) + \cos(x)$ on $[-5, 5]$ with networks of width 16, using different activation functions.

**Conclusion**   In this section, we explore the approximation performance of different activation functions for one hidden-layer network, both in theory and by experiment. We identify that the results of the experiments are nearly corresponded to the propositions in section 4.1. This section also reveals there may be potential correlation between representation and optimization. As we all know, normalization is crucial especially in deep networks. Since PLN and PLS can activate the deep neural networks, we will further explore what role normalization plays, in the following section.

## 5. Normalization or Activation?

Current deep neural networks usually consist of three parts: linear layers (store the parameters), nonlinear layers (usually the activation functions) and normalization (control the data distribution and stable training). Based on the preceding discussion, we pose the following question: Are both normalization and traditional activation functions (e.g. ReLU) necessary? We conduct experiments in different scenarios and attempt to answer this question.

### 5.1. PLN as Activation in DNNs

In this subsection, we investigate the performance of PLN as an activation function within both CNN and Transformer architectures.

#### 5.1.1. NETWORKS WITHOUT NORMALIZATION

We trace back to a past scenario, when normalization techniques had not been introduced into DNNs. One of the methods that improve training is weight initialization (He et al., 2015; LeCun et al., 2002) . Differently, our idea is

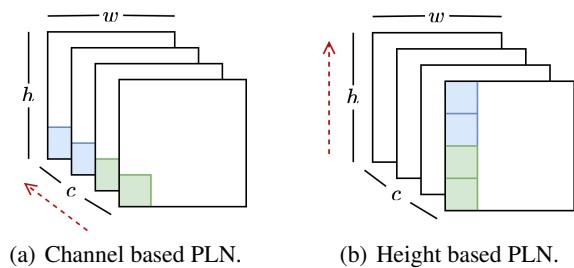

(a) Channel based PLN.  (b) Height based PLN.

*Figure 6.* The figures shows how Channel-PLN and Height-PLN compute on neurons, where we conduct LN within each region of the same color. Width-PLN can be similarly defined.

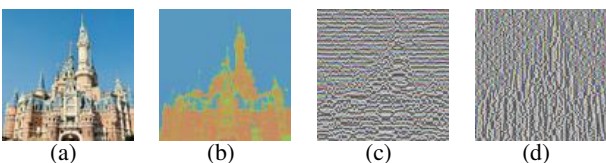

(a)  (b)  (c)  (d)

*Figure 7.* Figure (a) is the original image, and the size is 3×90×90. Figures (b), (c), and (d) are processed from Figure (a) by channel, height, and width based PLN-3, respectively. Among them, Figure (b), which uses Channel-PLN, seems retain more of the original information compared to Figures (c) and (d).

to replace the activation function, akin to the progression from sigmoid (McCulloch & Pitts, 1943) to tanh (Graves & Graves, 2012) and then ReLU (Krizhevsky et al., 2012). We conduct experiments on VGG, ResNet without BN, and Transformer without LN.

**Image Classification with CNN.**   To apply PLN on images, we design Channel-PLN. Channel-PLN calculates the mean/variance along only the channel dimension and use separate statistics over each position (a pair of height and width), as shown in Figure 6(a). In fact, we can also define Height-PLN (shown in Figure 6(b)) and Width-PLN. All of them are nonlinear layers, but Channel-PLN is the one we recommend and use in this paper, since it can retain more information of the original image after the normalization[2] as shown in Figure 7. Besides, Channel-PLN follows the calculation like MLP. The width in MLP is regarded equilent to the channels in CNN. Therefore, we use Channel-PLN as default, and note it PLN for simplification.

We apply the origin VGG structure (without Batch Normalization) in our experiments to compare with different activation functions (please see *Appendix* B.2.1 for the detailed experiment settings). In the meanwhile, we record the average norm of the gradient of the initial parameters. We recommend 8 as the norm size of PLN in CNN, please see *Appendix* B.2.2 for the experiments on norm sizes. We

---

[2]When we apply PLN on an image, we will get negative outputs. Therefore, we conduct a reversed-LN on the output, to ensure it has the same mean and variance as the origin photo, among all the pixel points.

*Table 1.* Results on VGG-16 using different activation functions.

| Activation | Train Acc(%) | Test Acc(%) | Gradient |
|---|---|---|---|
| PLN-8 | 88.76 | 89.45 | 0.0068 |
| Sigmoid | 9.81 | 10.00 | 0.0026 |
| Tanh | 9.76 | 10.00 | 0.0006 |
| ReLU | 9.76 | 10.00 | 0.0006 |

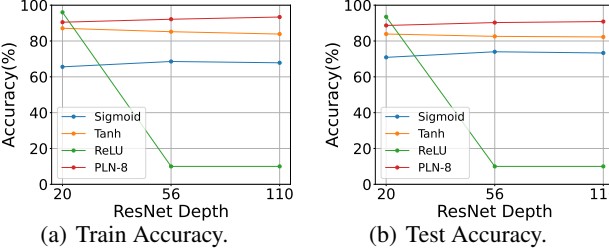

(a) Train Accuracy.    (b) Test Accuracy.

*Figure 8.* Results on ResNet of different depths without BN, using different activation functions.

show the results on VGG-16 in Table 1.

We find that the traditional activation functions are hard to train using origin VGG architectures. However, PLN-8 can keep its optimization property and perform well. By analyzing the gradients, we conjecture that gradient vanishing is probable the reason traditional activations do not work.

We further conduct experiments on ResNet architectures without BN, where the residual connection can avoid gradient vanishing. We change the learning rate to 0.01 on ResNet-20, ResNet-56 and ResNet-110. The other experiment settings are the same. Please see Figure 8 for the detailed results and Table 2 for the gradient information.

Different from the results of VGG, we find that we can easily train sigmoid and tanh in ResNet architectures. However, as the depth increases, ReLU becomes hard to train without normalization. We deduct ReLU suffers from gradient explosion in deep ResNet architectures withoutBN, according to the gradient norms in Table 2. As for PLN-8, it performs well in such settings, for its good property both in representation and optimization.

**Time-series Tasks.** We conducted sequence prediction experiments on the Traffic dataset, enhanced through data extension, using a Transformer architecture. Specifically, we extended the sequence length processed in a single step from 96 to 720, while adhering to the remaining configurations of the Time Series Benchmark (Wang et al., 2024).

We find that while PLN-16 does not outperform other methods in the Transformer architecture, it achieves comparable performance to ReLU. When no other normalization is present, PLN demonstrates the strongest optimization capability as an activation function. Furthermore, when PLN16 serves as a normalization layer, it achieves good performance even without activation functions.

*Table 2.* Initial gradients on ResNet without BN.

| Activation | ResNet-20 | ResNet-56 | ResNet-110 |
|---|---|---|---|
| Sigmoid | 0.13 | 0.533 | 170.3 |
| Tanh | 0.04 | 0.101 | 27.06 |
| ReLU | 1.68 | $2.5 \times 10^4$ | $1.2 \times 10^{13}$ |
| PLN-8 | 0.08 | 0.179 | 41.01 |

*Table 3.* Results on the Traffic Dataset using Transformer without LN. We record the MSE using different activation functions, where lower MSE indicates better performance.

| MSE | PLN-16 | ReLU | Tanh | GeLU | Sigmoid | Identity |
|---|---|---|---|---|---|---|
| Identity | 0.7391 | 0.7602 | 0.7716 | 0.7802 | 0.7939 | 0.7551 |

In this subsection, we conclude that PLN with proper norm size can perform well using only linear modules. This is because PLN shows good property both in representation and optimization.

### 5.1.2. NETWORKS WITH OTHER NORMALIZATIONS

We also conduct experiments by replacing the activation functions with PLN and PLS, in networks with normalizations. We fix the norm size $d = 8$ for PLN and PLS, while width ranges in $16, 32, 64, 128, 256$. Besides, we compare the performance with sigmoid, tanh and ReLU. We conduct experiments on CIFAR-10 using VGG-16 with BN. The results are shown in Figure 9.

We find that in VGG-16 with BN, PLN-8 performs better than sigmoid and tanh, but slightly worse than ReLU. We posit that BN provides a more substantial boost to the representation capacity of ReLU compared to sigmoid and tanh.This conclusion is supported by the findings in Section 4.1, which indicate that the representation capacity of sigmoid and tanh is not inferior to that of ReLU.

Although the representation capacity of PLN-8 is not particularly strong, it still performs better than sigmoid and tanh. The results indicate that the optimization property of an activation function is also important. Although PLN does not outperform ReLU, we believe that PLN holds potential as an activation function, given its ease of training in deep neural networks.

### 5.2. PLN as Normalization in Networks

Although PLN possesses strong representation capacity, it evolves from normalization in the final analysis. This prompts us to pose the following question: when PLN is used as a normalization method, do we still require activation functions? We conduct experiments to answer the question.

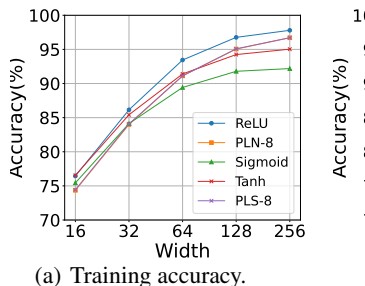
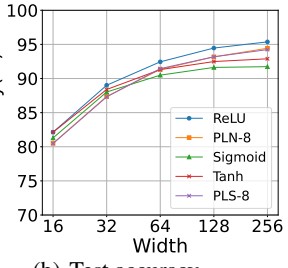

(a) Training accuracy.  (b) Test accuracy.

*Figure 9.* Results of different activation functions with different widths on CIFAR-10 using VGG-16 with BN.

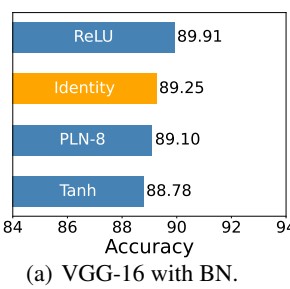
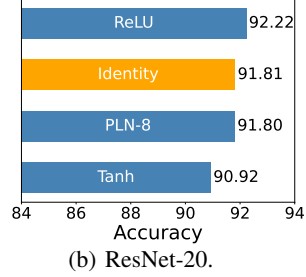

(a) VGG-16 with BN.  (b) ResNet-20.

*Figure 10.* The results using PLN as Normalization. The term "Identity" means there is "no activation function" in the network, as a reference.

### 5.2.1. REPLACE BN WITH PLN IN CNN

We follows the experiment settings in section 5.1.1 using PLN-8 as normalization rather than BN, with different activation functions. We record the test accuracy(%) on CIFAR-10 using VGG-16 and ResNet-20 in Figure 10.

When using PLN-8 for normalization, we observe that the accuracy improves only marginally with the addition of ReLU. Sigmoid and tanh even reduce the accuracy. This indicates that when normalization itself has strong representation power, extra activation functions might not be essential.

### 5.2.2. TRANSFORMER NORMALIZED BY PLN

We conduct experiments using Transformer on machine translation tasks. We employed the Transformer model and evaluated it on the Multi30K dataset (please see *Appendix B.3.1* for the detailed experiment settings).We compared the experimental results obtained using PLN-8 as the normalization method across various activation functions. The BLEU scores (where higher values indicate better performance) for the test set are shown as the orange columns in Figure 11. In contrast to the results in CNNs, we find that the use of GELU or ReLU leads to a substantial improvement in the model's performance relative to the Identity function.We also conduct experiments using the original normalization (LN). We find the results of using LN is worse than that of PLN-8. Given that PLN-8 exhibits stronger nonlinearity than LN, we conjecture that translation tasks demand greater

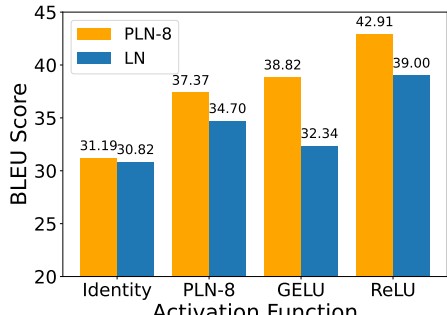

*Figure 11.* Test BLEU Score of Transformer on Multi30k.The orange histogram represents the use of PLN-8 as the normalization layer, while the blue histogram represents the use of LN as the normalization layer. The horizontal axis denotes different activation functions.

representation capacity. This may explain why introducing ReLU significantly enhances performance in networks with PLN as normalization layers.Besides, we figure out that the combination of PLN-8 and ReLU performs exceptionally well, achieving a score of 42.91.

## 6. Conclusion

We mathematically proved that a network with parallel layer normalizations (PLN) and linear layers only has universal approximation ability. We also theoretically measured the ability by discussing on approximating $L$-Lipchitz continuous functions. We also apply this measuring method for other activation functions (e.g., ReLU). We find that PLN has a little weaker representation capacity with sigmoid and ReLU, but stands out for its excellent optimization property as normalization itself. We believe it meaningful to research on the optimization property of activation functions, and even any nonlinear layers in neuron networks.

**Limitation and Future Work.** The effectiveness of parallel layer normalizations (PLN) is only verified on small-scale networks and datasets, and more results on large-scale networks and datasets are required to support the practicality of PLN. There are much empirical tricks on training a network, but it may be not suitable for a network without traditional activation functions. We have not fully utilized the potential capabilities of PLN. Nevertheless, we still believe the combination of representation and optimization in PLN will refresh and improve our understandings of DNNs.

## Impact Statement

This paper presents work whose goal is to advance the field of Machine Learning. There are many potential social consequences of our work, none of which feels it must be specifically highlighted here.

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

## A. Mathematical Proofs

### A.1. Proof of Lemma 1 at the case $d > 2$

**Lemma 1.** *There is a $G(\mathbf{x}) \in \mathcal{G}(N+1; LN)$, subjected to that $G(\mathbf{x})$ is a linear combination with $N$ bounded measurable sigmoidal functions.*

*Proof.* Here we give the proof at the case $d > 2$.

Assume that $G(\mathbf{x}) = \sum_{j=1}^{N+1} \boldsymbol{\alpha}_j^\top LN(\boldsymbol{W}_j\mathbf{x} + \boldsymbol{b}_j)$. For $1 \leq j \leq N$, let $\boldsymbol{\alpha}_j = [\hat{\alpha}_j, 0, 0, \cdots, 0]^\top, \boldsymbol{W}_j = [\boldsymbol{w}_j, -\boldsymbol{w}_j, \mathbf{0}, \cdots, \mathbf{0}]^\top$

and $\boldsymbol{b}_j = [b_j, -b_j, 0, \cdots, 0]^\top$. Let $\boldsymbol{\alpha}_{N+1} = [(\hat{\alpha}_1 + \cdots + \hat{\alpha}_N), 0, 0, \cdots, 0]^\top, \boldsymbol{W}_{N+1} = \boldsymbol{O}$ and $\boldsymbol{b}_{N+1} = [1, -1, 0, \cdots, 0]^\top$. According to Eqn.3, it is easy to identify that for $1 \leq j \leq N, \boldsymbol{W}_j\mathbf{x} + \boldsymbol{b}_j = [\boldsymbol{w}_j^\top\mathbf{x} + b_j, -(\boldsymbol{w}_j^\top\mathbf{x} + b_j), 0, \cdots, 0]^\top$, while $\boldsymbol{W}_{N+1}\mathbf{x} + \mathbf{b}_{N+1} = [1, -1, 0, \cdots, 0]^\top$. Here we have:

$$
\begin{aligned}
G(\mathbf{x}) &= \sum_{j=1}^{N} \hat{\alpha}_j \cdot \frac{\boldsymbol{w}_j^\top\mathbf{x} + b_j}{\sqrt{\frac{2}{d}(\boldsymbol{w}_j^\top\mathbf{x} + b_j)^2}} + \sum_{j=1}^{N} \sqrt{\frac{d}{2}}\hat{\alpha}_j \\
&= \sum_{j=1}^{N} \sqrt{2d}\hat{\alpha}_j \left[ \frac{\boldsymbol{w}_j^\top\mathbf{x} + b_j}{2|\boldsymbol{w}_j^\top\mathbf{x} + b_j|} + \frac{1}{2} \right] \\
&= \sum_{j=1}^{N} \sqrt{2d}\hat{\alpha}_j \sigma(\boldsymbol{w}_j^\top\mathbf{x} + b_j),
\end{aligned}
\tag{11}
$$

where $\sigma(x) = (x/|x|+1)/2$ is obvious a bounded measurable sigmoidal function, even though it is not defined at $x = 0$. $\quad\square$

### A.2. Proof of Corollary 2

**Corallary 2.** (LS for Universal Approximation Theorem.) *Let $LS(\cdot)$ be Layer Scaling (i.e. RMSNorm) on $\mathbb{R}^d, d \geq 1$. Given any $f \in C([0,1]^n)$ and $\varepsilon > 0$, there is a sum $G(\mathbf{x}) \in \mathcal{G}(N; LS)$ when $N$ is large enough, subjected to $|G(\mathbf{x}) - f(\mathbf{x})| < \varepsilon$ for $\mathbf{x} \in [0,1]^n$.*

*Proof.* The proof is similar to that of LN. Assume that $G(\mathbf{x}) = \sum_{j=1}^{N+1} \boldsymbol{\alpha}_j^\top LS(\boldsymbol{W}_j\mathbf{x} + \boldsymbol{b}_j)$. For $1 \leq j \leq N$, let $\boldsymbol{\alpha}_j = [\hat{\alpha}_j, 0, \cdots, 0]^\top, \boldsymbol{W}_j = [\boldsymbol{w}_j, \mathbf{0}, \cdots, \mathbf{0}]^\top$ and $\boldsymbol{b}_j = [b_j, 0, \cdots, 0]^\top$. Let $\boldsymbol{\alpha}_{N+1} = [(\hat{\alpha}_1 + \cdots + \hat{\alpha}_N), 0, \cdots, 0]^\top, \boldsymbol{W}_{N+1} = \boldsymbol{O}$ and $\boldsymbol{b}_{N+1} = [1, 0, \cdots, 0]^\top$. According to Eqn.6, it is easy to identify that for $1 \leq j \leq N, \boldsymbol{W}_j\mathbf{x} + \boldsymbol{b}_j = [\boldsymbol{w}_j^\top\mathbf{x} + b_j, 0, \cdots, 0]^\top$ while $\boldsymbol{W}_{N+1}\mathbf{x} + \boldsymbol{b}_{N+1} = [1, 0, \cdots, 0]^\top$. Here we have:

$$
\begin{aligned}
G(\mathbf{x}) &= \sum_{j=1}^{N} \hat{\alpha}_j \cdot \frac{\boldsymbol{w}_j^\top\mathbf{x} + b_j}{\sqrt{\frac{1}{d}(\boldsymbol{w}_j^\top\mathbf{x} + b_j)^2}} + \sum_{j=1}^{N} \sqrt{d}\hat{\alpha}_j \\
&= \sum_{j=1}^{N} 2\sqrt{d}\hat{\alpha}_j \left[ \frac{\boldsymbol{w}_j^\top\mathbf{x} + b_j}{2|\boldsymbol{w}_j^\top\mathbf{x} + b_j|} + \frac{1}{2} \right] \\
&= \sum_{j=1}^{N} 2\sqrt{d}\hat{\alpha}_j \sigma(\boldsymbol{w}_j^\top\mathbf{x} + b_j),
\end{aligned}
\tag{12}
$$

where $\sigma(x) = (x/|x| + 1)/2$ is obvious a bounded measurable sigmoidal function, even though it is not defined at $x = 0$.

Furthermore, by Theorem 1, we can prove Corallary 2. $\quad\square$

### A.3. Proof of Proposition 1

**Proposition 1.** (Approximation Bound of LN)

Given $\mathcal{F} = \mathcal{F}([0,1]; L)$ and $\mathcal{G} = \mathcal{G}(N; LN)$, where $LN(\cdot)$ denotes LN on $\mathbb{R}^d$, $d \geq 2$. Given the error bound $\varepsilon > 0$, we have

$$\mathcal{N}(LN) \leq \lfloor L/2\varepsilon \rfloor + 1. \tag{13}$$

Furthermore, we have $\mathcal{W}(LN) \leq 2(\lfloor L/2\varepsilon \rfloor + 1)$.

Here, we consider the case there is a small number $\delta > 0$ in practical LN. $\delta$ is a small number for numerical stability in LN. Specifically, we rewrite Eqn.3 as

$$\hat{x}_j = \frac{x_j - \mu}{\sigma + \delta}. \tag{14}$$

In the following section, we first prove Lemma 2 and Lemma 3 and then proceed with the formal proof.

## A.3.1. REQUIRED LEMMAS

**Lemma 2.** *Given a $\hat{G}(x) \in \mathcal{G}(N; \mathrm{sign})$, there is a $G(x) \in \mathcal{G}(N; LN)$, subjected to $\lim_{\delta \to 0^+} G(x) = \hat{G}(x)$. Here $\mathrm{sign}(x)$ is the sign function, which outputs $-1, 0, 1$ when $x < 0, x = 0, x > 0$ respectively.*

*Proof.* Assume $G(x) = \sum_{j=1}^{N} \boldsymbol{\alpha}_j^\top LN(\boldsymbol{w}_j x + \boldsymbol{b}_j)$. Let $\boldsymbol{\alpha}_j = [\hat{\alpha}_j \sqrt{2/d}, 0, \cdots, 0]^\top$, $\boldsymbol{w}_j = [\hat{w}_j, -\hat{w}_j, 0, \cdots, 0]^\top$ and $\boldsymbol{b}_j = [\hat{b}, -\hat{b}_j, 0, \cdots, 0]^\top$, for $1 \leq j \leq N$. It is easy to identify that:

$$\lim_{\delta \to 0^+} \boldsymbol{\alpha}_j^\top LN(\boldsymbol{w}_j x + \boldsymbol{b}_j) = \lim_{\delta \to 0^+} \hat{\alpha}_j \sqrt{2/d} \cdot \frac{\hat{w}_j x + \hat{b}_j}{\sqrt{2(\hat{w}_j x + \hat{b}_j)^2/d + \delta}}$$

$$= \lim_{\delta \to 0^+} \frac{\hat{\alpha}_j(\hat{w}_j x + \hat{b}_j)}{|\hat{w}_j x + \hat{b}_j| + \delta\sqrt{d/2}} \tag{15}$$

$$= \hat{\alpha}_j \, \mathrm{sign}(\hat{w}_j x + \hat{b}_j),$$

even if $\hat{w}_j x + \hat{b}_j = 0$.

Given $\hat{G}(x) \in \mathcal{G}(N; \mathrm{sign})$, we have:

$$\hat{G}(x) = \sum_{j=1}^{N} \hat{\alpha}_j \, \mathrm{sign}(\hat{w}_j x + \hat{b}_j)$$

$$= \lim_{\delta \to 0^+} \sum_{j=1}^{N} \boldsymbol{\alpha}_j^\top LN(\boldsymbol{w}_j x + \boldsymbol{b}_j) \tag{16}$$

$$= \lim_{\delta \to 0^+} G(x),$$

where $\boldsymbol{\alpha}_j, \boldsymbol{w}_j$ and $\boldsymbol{b}_j$ can be determined by $\hat{\alpha}_j, \hat{w}_j, \hat{b}_j$ for each $j$.

Therefore, we have the conclusion that there is a $G(x) \in \mathcal{G}(N; LN)$, subjected to $\lim_{\delta \to 0^+} G(x) = \hat{G}(x)$. $\square$

**Lemma 3.** *Given any L-Lipschitz continuous function $f \in [0,1]$ and the error $\varepsilon > 0$, there is some $\hat{G}(x) \in \mathcal{G}(\lfloor L/2\varepsilon \rfloor + 1; \mathrm{sign})$, subjected to $|\hat{G}(x) - f(x)| < \varepsilon$ for $x \in [0,1]$.*

*Proof.* Given $\hat{G}(x) = \sum_{j=1}^{N} \hat{\alpha}_j \, \mathrm{sign}(\hat{w}_j x + \hat{b}_j)$, where $N = \lfloor L/2\varepsilon \rfloor + 1$. For $1 \leq j \leq N-1$, we set

$$\hat{\alpha}_j = \frac{1}{2}\left[ f\left(\frac{2j+1}{2N}\right) - f\left(\frac{2j-1}{2N}\right) \right]; \tag{17}$$

while

$$\hat{\alpha}_N = \frac{1}{2}\left[ f\left(\frac{1}{2N}\right) + f\left(\frac{2N-1}{2N}\right) \right]. \tag{18}$$

Besides, we set $\hat{w}_j = 1$ for $1 \leq j \leq N$, $\hat{b}_j = -\dfrac{j}{N}$ for $1 \leq j \leq N-1$, and $\hat{b}_N = 1$.

This case, for $\dfrac{j-1}{N} < x < \dfrac{j}{N}$ where $1 \leq j \leq N$, we obtain that:

$$
\begin{aligned}
\hat{G}(x) &= \sum_{k=1}^{j-1} \hat{\alpha}_k - \sum_{k=j}^{N-1} \hat{\alpha}_k + \hat{\alpha}_N \\
&= \frac{1}{2}\left[ f\left(\frac{2j-1}{2N}\right) - f\left(\frac{1}{2N}\right) \right] - \frac{1}{2}\left[ f\left(\frac{2N-1}{2N}\right) - f\left(\frac{2j-1}{2N}\right) \right] + \frac{1}{2}\left[ f\left(\frac{1}{2N}\right) + f\left(\frac{2N-1}{2N}\right) \right] \quad (19) \\
&= f\left(\frac{2j-1}{2N}\right).
\end{aligned}
$$

As for $x = \dfrac{j}{N}$ where $1 \leq j \leq N-1$, we have:

$$
\begin{aligned}
\hat{G}(x) &= \sum_{k=1}^{j-1} \hat{\alpha}_k - \sum_{k=j+1}^{N-1} \hat{\alpha}_k + \hat{\alpha}_N \\
&= \frac{1}{2}\left[ f\left(\frac{2j-1}{2N}\right) - f\left(\frac{1}{2N}\right) \right] - \frac{1}{2}\left[ f\left(\frac{2N-1}{2N}\right) - f\left(\frac{2j+1}{2N}\right) \right] + \frac{1}{2}\left[ f\left(\frac{1}{2N}\right) + f\left(\frac{2N-1}{2N}\right) \right] \quad (20) \\
&= \frac{1}{2}\left[ f\left(\frac{2j-1}{2N}\right) + f\left(\frac{2j+1}{2N}\right) \right].
\end{aligned}
$$

Besides, we have $\hat{G}(0) = f\left(\dfrac{1}{2N}\right)$, and $\hat{G}(1) = f\left(\dfrac{2N-1}{2N}\right)$.

Since $\lfloor L/2\varepsilon \rfloor \leq L/2\varepsilon < \lfloor L/2\varepsilon \rfloor + 1$, we have $N > L/2\varepsilon$. Then we obtain that:

1) If $x = 0$, we have:

$$
\begin{aligned}
|\hat{G}(0) - f(0)| &= \left| f(0) - f\left(\frac{1}{2N}\right) \right| \\
&\leq \frac{L}{2N} \quad (21) \\
&< \varepsilon.
\end{aligned}
$$

2) If $x = 1$, we have:

$$
\begin{aligned}
|\hat{G}(1) - f(1)| &= \left| f(1) - f\left(\frac{2N-1}{2N}\right) \right| \\
&\leq \frac{L}{2N} \quad (22) \\
&< \varepsilon.
\end{aligned}
$$

3) If $x = \dfrac{j}{N}$, we have:

$$
\begin{aligned}
\left| \hat{G}\left(\frac{j}{N}\right) - f\left(\frac{j}{N}\right) \right| &= \left| \frac{1}{2}\left[ f\left(\frac{2j-1}{2N}\right) + f\left(\frac{2j+1}{2N}\right) \right] - f\left(\frac{j}{N}\right) \right| \\
&\leq \frac{1}{2}\left| f\left(\frac{2j-1}{2N}\right) - f\left(\frac{j}{N}\right) \right| + \frac{1}{2}\left| f\left(\frac{2j+1}{2N}\right) - f\left(\frac{j}{N}\right) \right| \quad (23) \\
&\leq \frac{L}{4N} + \frac{L}{4N} \\
&< \varepsilon.
\end{aligned}
$$

4) If $\dfrac{j-1}{N} < x < \dfrac{j}{N}$, we have:

$$
\begin{aligned}
\left| \hat{G}(x) - f(x) \right| &= \left| f\left( \frac{2j-1}{2N} \right) - f(x) \right| \\
&\leq L \left| \frac{2j-1}{2N} - x \right| \\
&< \frac{L}{2N} \\
&< \varepsilon.
\end{aligned}
\tag{24}
$$

Therefore, for $x \in [0,1]$ belongs to one of the four cases above, fulfilling $|\hat{G}(x) - f(x)| < \varepsilon$. $\qquad\square$

### A.3.2. FORMAL PROOF.

*Proof.* We prove Proposition A.3 based on the proof above.

According to the proof of Lemma 2, we denote that $G(x) = \sum_{j=1}^{N} \hat{\alpha}_j s_j(x)$, where $s_j(x) = \dfrac{x + \hat{b}_j}{|x + \hat{b}_j| + \delta}$ and $\delta > 0$ is the small number in LN for numerical stability.

In the proof here, based on Eqn.15, we simplify $\delta\sqrt{d/2}$ as $\delta$, since they are almost the same for $\delta \to 0$. On the other hand, this simplification can be also seen as the proof of the case $d = 2$, which is easy to extend to $d > 2$.

Next, we discuss $|\hat{G}(x) - G(x)|$ for $x \in [0,1]$ in the following two cases. We set $\delta_0 \in \left( 0, \dfrac{1}{2N} \right)$.

1) If $x$ satisfies: $\forall j = 1, 2, \cdots, N$, we have $|x + \hat{b}_j| > \delta_0$. Based on the proof of Lemma 3, we have $|\hat{G}(x) - f(x)| < \varepsilon$. Furthermore, there is some $\varepsilon_1 > 0$, subjected to $|\hat{G}(x) - f(x)| \leq \varepsilon - \varepsilon_1$. Here we obtain:

$$
\begin{aligned}
|\hat{G}(x) - G(x)| &= \left| \sum_{j=1}^{N} \hat{\alpha}_j \operatorname{sign}(x + \hat{b}_j) - \sum_{j=1}^{N} \hat{\alpha}_j \frac{x + \hat{b}_j}{|x + \hat{b}_j| + \delta} \right| \\
&= \left| \sum_{j=1}^{N} \hat{\alpha}_j \left[ \frac{x + \hat{b}_j}{|x + \hat{b}_j|} - \frac{x + \hat{b}_j}{|x + \hat{b}_j| + \delta} \right] \right| \\
&\leq \sum_{j=1}^{N} |\hat{\alpha}_j| \left| \frac{x + \hat{b}_j}{|x + \hat{b}_j|} - \frac{x + \hat{b}_j}{|x + \hat{b}_j| + \delta} \right| \\
&= \sum_{j=1}^{N} |\hat{\alpha}_j| \left| \frac{\delta(x + \hat{b}_j)}{|x + \hat{b}_j|(|x + \hat{b}_j| + \delta)} \right| \\
&= \sum_{j=1}^{N} |\hat{\alpha}_j| \cdot \frac{\delta}{|x + \hat{b}_j| + \delta}.
\end{aligned}
\tag{25}
$$

Given $\alpha^* = \max_{1 \leq j \leq N} |\hat{\alpha}_j|$ and $\delta_N = \dfrac{\varepsilon_1 \delta_0}{N\alpha^*}$, for $\delta \leq \delta_N$, we have:

$$
\begin{aligned}
|\hat{G}(x) - G(x)| &\leq \sum_{j=1}^{N} |\hat{\alpha}_j| \cdot \frac{\delta}{|x + \hat{b}_j| + \delta} \\
&< \sum_{j=1}^{N} \alpha^* \cdot \frac{\varepsilon_1 \delta_0}{N\alpha^*} \cdot \frac{1}{\delta_0 + \delta} \\
&= \frac{\varepsilon_1 \delta_0}{\delta_0 + \delta} \\
&< \varepsilon_1.
\end{aligned}
\tag{26}
$$

Therefore, we have:

$$|G(x) - f(x)| \leq |G(x) - \hat{G}(x)| + |\hat{G}(x) - f(x)|$$
$$< \varepsilon_1 + \varepsilon - \varepsilon_1 \tag{27}$$
$$= \varepsilon.$$

2) If there exists some $k$ that satisfied $|x + \hat{b}_k| \leq \delta_0$—for $x \in [0, 1]$ and $\delta_0 \in \left(0, \frac{1}{2N}\right)$, we have $1 \leq k \leq N - 1$ and $\hat{b}_k = -\frac{k}{N}$. Since $N = \lfloor L/2\varepsilon \rfloor + 1$, we have $N > \frac{L}{2\varepsilon}$. Hence, there is some $\varepsilon_2 > 0$, subjected to that $\frac{L}{2N} \leq \varepsilon - \varepsilon_2$. Here we rewrite:

$$|G(x) - f(x)| = |G(x) - f(x) + \hat{G}(x) - \hat{G}(x)|$$

$$= \left| \sum_{j \neq k} \hat{\alpha}_j s_j(x) + \hat{\alpha}_k s_k(x) - f(x) + \hat{G}(x) - \sum_{j \neq k} \hat{\alpha}_j \text{sign}(x + \hat{b}_j) - \hat{\alpha}_k \text{sign}(x + \hat{b}_k) \right| \tag{28}$$

$$\leq \left| \sum_{j \neq k} \hat{\alpha}_j s_j(x) - \sum_{j \neq k} \hat{\alpha}_j \text{sign}(x + \hat{b}_j) \right| + \left| \hat{G}(x) + \hat{\alpha}_k s_k(x) - \hat{\alpha}_k \text{sign}(x + \hat{b}_k) - f(x) \right|.$$

For the first term, similar to case 1, we set $\alpha_k^* = \max_{j \neq k} |\hat{\alpha}_j|$ and $\delta_k = \frac{\varepsilon_2 \delta_0}{(N-1)\alpha_k^*}$. For $\delta \leq \delta_k$, we have:

$$\left| \sum_{j \neq k} \hat{\alpha}_j s_j(x) - \sum_{j \neq k} \hat{\alpha}_j \text{sign}(x + \hat{b}_j) \right|$$

$$= \left| \sum_{j \neq k} \hat{\alpha}_j \left[ \frac{x + \hat{b}_j}{|x + \hat{b}_j| + \delta} - \frac{x + \hat{b}_j}{|x + \hat{b}_j|} \right] \right|$$

$$\leq \sum_{j \neq k} |\hat{\alpha}_j| \left| \frac{x + \hat{b}_j}{|x + \hat{b}_j| + \delta} - \frac{x + \hat{b}_j}{|x + \hat{b}_j|} \right|$$

$$= \sum_{j \neq k} |\hat{\alpha}_j| \left| \frac{\delta(x + \hat{b}_j)}{|x + \hat{b}_j|(|x + \hat{b}_j| + \delta)} \right| \tag{29}$$

$$= \sum_{j \neq k} |\hat{\alpha}_j| \cdot \frac{\delta}{|x + \hat{b}_j| + \delta}$$

$$< \sum_{j \neq k} \alpha_k^* \cdot \frac{\varepsilon_2 \delta_0}{(N-1)\alpha_k^*} \cdot \frac{1}{\delta_0 + \delta}$$

$$= \frac{\varepsilon_2 \delta_0}{\delta_0 + \delta}$$

$$< \varepsilon_2.$$

For the second term, notice that when $\frac{k}{N} - \delta_0 \leq x \leq \frac{k}{N} + \delta_0$, we have

$$\hat{G}(x) = \begin{cases} f\left(\frac{2k-1}{2N}\right), & \frac{k}{N} - \delta_0 \leq x < \frac{k}{N} \\ \frac{1}{2}f\left(\frac{2k-1}{2N}\right) + \frac{1}{2}f\left(\frac{2k+1}{2N}\right), & x = \frac{k}{N} \\ f\left(\frac{2k+1}{2N}\right), & \frac{k}{N} < x \leq \frac{k}{N} + \delta_0, \end{cases} \tag{30}$$

and

$$\hat{\alpha}_k \text{sign}(x + \hat{b}_k) = \begin{cases} \frac{1}{2}\left[f\left(\frac{2k-1}{2N}\right) - f\left(\frac{2k+1}{2N}\right)\right], & \frac{k}{N} - \delta_0 \leq x < \frac{k}{N} \\ 0, & x = \frac{k}{N} \\ \frac{1}{2}\left[f\left(\frac{2k+1}{2N}\right) - f\left(\frac{2k-1}{2N}\right)\right], & \frac{k}{N} < x \leq \frac{k}{N} + \delta_0. \end{cases} \tag{31}$$

We thus have

$$\hat{G}(x) - \hat{\alpha}_k \, \text{sign}(x + \hat{b}_k) = \frac{1}{2}\left[f\left(\frac{2k+1}{2N}\right) + f\left(\frac{2k-1}{2N}\right)\right], \quad \text{for } \frac{k}{N} - \delta_0 \leq x \leq \frac{k}{N} + \delta_0. \tag{32}$$

As for

$$\hat{\alpha}_k s_k(x) = \frac{1}{2}\left[f\left(\frac{2k+1}{2N}\right) - f\left(\frac{2k-1}{2N}\right)\right] \cdot \frac{x - \frac{k}{N}}{|x - \frac{k}{N}| + \delta}, \tag{33}$$

since $s_k(x) \in (-1, 1)$ and $s_k(x)\left(\frac{k}{N} - x\right) \leq 0$, we obtain that:

$$\begin{aligned}
&\left|\hat{G}(x) - \hat{\alpha}_k\text{sign}(x + \hat{b}_k) + \hat{\alpha}_k s_k(x) - f(x)\right| \\
&= \left|\frac{1 + s_k(x)}{2}f\left(\frac{2k+1}{2N}\right) + \frac{1 - s_k(x)}{2}f\left(\frac{2k-1}{2N}\right) - f(x)\right| \\
&\leq \left|\frac{1 + s_k(x)}{2}\right|\left|f\left(\frac{2k+1}{2N}\right) - f(x)\right| + \left|\frac{1 - s_k(x)}{2}\right|\left|f\left(\frac{2k-1}{2N}\right) - f(x)\right| \\
&\leq \frac{1 + s_k(x)}{2} \cdot L \cdot \left(\frac{2k+1}{2N} - x\right) + \frac{1 - s_k(x)}{2} \cdot L \cdot \left(x - \frac{2k-1}{2N}\right) \\
&= \frac{1}{2}L \cdot \frac{1}{N} + \frac{s_k(x)}{2} \cdot L \cdot \left(\frac{2k+1}{2N} - x\right) + \frac{s_k(x)}{2} \cdot L \cdot \left(\frac{2k-1}{2N} - x\right) \\
&= \frac{L}{2N} + Ls_k(x)\left(\frac{k}{N} - x\right) \\
&\leq \frac{L}{2N} \\
&\leq \varepsilon - \varepsilon_2.
\end{aligned} \tag{34}$$

Accordingly, we have:

$$\begin{aligned}
|G(x) - f(x)| &\leq \left|\sum_{j \neq k}\hat{\alpha}_j s_j(x) - \sum_{j \neq k}\hat{\alpha}_j \, \text{sign}(x + \hat{b}_j)\right| + \left|\hat{G}(x) + \hat{\alpha}_k s_k(x) - \hat{\alpha}_k\text{sign}(x + \hat{b}_k) - f(x)\right| \\
&< \varepsilon_2 + \varepsilon - \varepsilon_2 \\
&= \varepsilon.
\end{aligned} \tag{35}$$

Therefore, given $\delta^* = \min(\delta_1, \delta_2, \cdots, \delta_N)$, when $\delta \leq \delta^*$, we have $|G(x) - f(x)| < \varepsilon, \forall x \in [0, 1]$.

Consequently, we have proved that $\mathcal{N}(LN) \leq \lfloor L/2\varepsilon \rfloor + 1$ and $\mathcal{W}(LN) \leq 2(\lfloor L/2\varepsilon \rfloor + 1)$. $\square$

### A.4. Proof of Proposition 2

**Proposition 2.** (Approximation Bound of Sigmoid) *Given $\mathcal{F} = \mathcal{F}([0, 1]; L)$ and $\mathcal{G} = \mathcal{G}(N; \sigma)$, where $\sigma(x) = 1/(1 + e^{-x})$ denotes the sigmoid function. Given the error bound $\varepsilon > 0$, we have*

$$\mathcal{N}(\sigma) \leq \lfloor L/2\varepsilon \rfloor + 1. \tag{36}$$

*Furthermore, we have $\mathcal{W}(\sigma) \leq 2(\lfloor L/2\varepsilon \rfloor + 1)$.*

### A.4.1. SIGMOID

We give the similar proof: we use sign as a bridge of our proof, with limitation notation. Then the idea of the proof is almost the same as LN. Here is the proof.

*Proof.* We denote $G(x) \in \mathcal{G}(N; \sigma)$ as $G(x) = \sum_{j=1}^{N} \alpha_j \sigma(w_j x + b_j)$, specialized as $G(x) = \sum_{j=1}^{N} \alpha_j \sigma[\lambda(x + b_j)]$, where $\sigma(x) = 1/(1 + e^{-x})$. Here we have:

$$
\begin{aligned}
\lim_{\lambda \to +\infty} G(x) &= \lim_{\lambda \to +\infty} \sum_{j=1}^{N} \alpha_j \sigma[\lambda(x + b_j)] \\
&= \sum_{j=1}^{N} \frac{1}{2} \alpha_j [\text{sign}(x + b_j) + 1].
\end{aligned}
\tag{37}
$$

Similarly, let $b_j = -\dfrac{j}{N}$ for $1 \le j \le N - 1$, and $b_N = 1$. We have:

$$
\lim_{\lambda \to +\infty} G(x) = \sum_{j=1}^{N-1} \frac{1}{2} \alpha_j \, \text{sign}(x + b_j) + \sum_{j=1}^{N-1} \frac{1}{2} \alpha_j + \alpha_N.
\tag{38}
$$

Let

$$
\alpha_j = f\left(\frac{2j+1}{2N}\right) - f\left(\frac{2j-1}{2N}\right),
\tag{39}
$$

for $1 \le j \le N - 1$, and $\alpha_N = f\left(\dfrac{1}{2N}\right)$.

Similar to the proof of Lemma 3, we obtain that $|\lim_{\lambda \to +\infty} G(x) - f(x)| < \varepsilon$ in $[0, 1]$.

Furthermore, with almost the same method of proving Proposition A.4, we can prove that $G(x) - f(x) < \varepsilon$.

With the two above conclusion, we can finish the proof. In the proof of Proposition A.4, $s_j(x)$ in Eqn.34 denotes $\dfrac{x + \hat{b}_j}{|x + \hat{b}_j| + \delta}$, while $s_j(x)$ here denotes $2\sigma[\lambda(x + b_j)] - 1 = \dfrac{1 - e^{-\lambda(x+b_j)}}{1 + e^{-\lambda(x+b_j)}}$, such that

$$
\lim_{\lambda \to +\infty} \sum_{j=1}^{N} \alpha_j s_j(x) = \sum_{j=1}^{N} \text{sign}(x + b_j).
\tag{40}
$$

Similarly, we consider two cases upon $x$:

1) If $x$ satisfies: $\forall j = 1, 2, \cdots, N$, we have $|x + \hat{b}_j| > \delta_0 > 0$. We also transfer $|\hat{G}(x) - f(x)| < \varepsilon$ to $|\hat{G}(x) - f(x)| \le \varepsilon - \varepsilon_1$.

Following Eqn.25, we replace with the new $s_j$, we have:

$$
\begin{aligned}
|\hat{G}(x) - G(x)| &= \left| \sum_{j=1}^{N} \frac{1}{2} [\alpha_j \text{sign}(x + b_j) + 1] - \sum_{j=1}^{N} \alpha_j \sigma[\lambda(x + b_j)] \right| \\
&= \sum_{j=1}^{N} \left| \frac{1}{2} \alpha_j [\text{sign}(x + b_j) - s_j(x)] \right| \\
&\le \sum_{j=1}^{N} \frac{1}{2} |\alpha_j| \left| \frac{x + b_j}{|x + b_j|} - \frac{1 - e^{-\lambda(x+b_j)}}{1 + e^{-\lambda(x+b_j)}} \right| \\
&\le \sum_{j=1}^{N} \frac{1}{2} \alpha^* \left| \frac{(x + b_j) + |x + b_j|}{|x + b_j|} - \frac{2}{1 + e^{-\lambda(x+b_j)}} \right|.
\end{aligned}
\tag{41}
$$

One different thing to do is to discuss the cases $x + b_j > \delta_0$ and $x + b_j < -\delta_0$ separately. Let $\lambda_N = \dfrac{1}{\delta_0} \ln \dfrac{N\alpha^* - \varepsilon_1}{\varepsilon_1}$ where $\alpha^* = \max\limits_{1 \le j \le N} |\hat{\alpha}_j|$. Here we will show that $|\hat{G}(x) - G(x)| < \varepsilon_1$ for $\lambda \ge \lambda_N$.

1.1) for the case $x + b_j > \delta_0 > 0$, we have:

$$
\begin{aligned}
&\left| \frac{(x + b_j) + |x + b_j|}{|x + b_j|} - \frac{2}{1 + e^{-\lambda(x + b_j)}} \right| \\
&= \left| 2 - \frac{2}{1 + e^{-\lambda(x + b_j)}} \right| \\
&= 2 - \frac{2}{1 + e^{-\lambda(x + b_j)}} \\
&< 2 - \frac{2}{1 + e^{-\lambda_N \delta_0}} \\
&= 2 - \frac{2}{1 + e^{\ln[\varepsilon_1/(N\alpha^* - \varepsilon_1)]}} \\
&= 2 - \frac{2(N\alpha^* - \varepsilon_1)}{N\alpha^* - \varepsilon_1 + \varepsilon_1} \\
&= \frac{2\varepsilon_1}{N\alpha^*}.
\end{aligned}
\tag{42}
$$

1.2) for the case $x + b_j < -\delta_0 < 0$, we have:

$$
\begin{aligned}
&\left| \frac{(x + b_j) + |x + b_j|}{|x + b_j|} - \frac{2}{1 + e^{-\lambda(x + b_j)}} \right| \\
&= \left| -\frac{2}{1 + e^{-\lambda(x + b_j)}} \right| \\
&= \frac{2}{1 + e^{-\lambda(x + b_j)}} \\
&< \frac{2}{1 + e^{\lambda_N \delta_0}} \\
&= \frac{2}{1 + e^{\ln[(N\alpha^* - \varepsilon_1)/\varepsilon_1]}} \\
&= \frac{2\varepsilon_1}{N\alpha^* - \varepsilon_1 + \varepsilon_1} \\
&= \frac{2\varepsilon_1}{N\alpha^*}.
\end{aligned}
\tag{43}
$$

Then, we have:

$$
\begin{aligned}
|\hat{G}(x) - G(x)| &\le \sum_{j=1}^{N} \frac{1}{2} \alpha^* \left| \frac{(x + b_j) + |x + b_j|}{|x + b_j|} - \frac{2}{1 + e^{-\lambda(x + b_j)}} \right| \\
&< \sum_{j=1}^{N} \frac{1}{2} \alpha^* \cdot \frac{2\varepsilon_1}{N\alpha^*} \\
&= \varepsilon_1.
\end{aligned}
\tag{44}
$$

Therefore, we have:

$$
\begin{aligned}
|G(x) - f(x)| &\le |G(x) - \hat{G}(x)| + |\hat{G}(x) - f(x)| \\
&< \varepsilon_1 + \varepsilon - \varepsilon_1 \\
&= \varepsilon.
\end{aligned}
\tag{45}
$$

2) If there exists some $k$ that satisfied $|x + \hat{b}_k| \leq \delta_0$—for $x \in [0, 1]$ and $\delta_0 \in \left(0, \dfrac{1}{2N}\right)$, we have $1 \leq k \leq N - 1$ and $\hat{b}_k = -\dfrac{k}{N}$. Similarly, here we construct $\dfrac{L}{2N} \leq \varepsilon - \varepsilon_2$ also. We can rewrite:

$$
\begin{aligned}
|G(x) - f(x)| &= |G(x) - f(x) + \hat{G}(x) - \hat{G}(x)| \\
&= \left| \sum_{j=1}^{N} \frac{1}{2} \alpha_j [s_j(x) + 1] - f(x) + \hat{G}(x) - \sum_{j=1}^{N} \frac{1}{2} \alpha_j [\mathrm{sign}(x + b_j) + 1] \right| \\
&\leq \frac{1}{2} \left| \sum_{j \neq k} \alpha_j s_j(x) - \sum_{j \neq k} \alpha_j \mathrm{sign}(x + b_j) \right| + \left| \hat{G}(x) + \frac{1}{2} \alpha_k s_k(x) - \frac{1}{2} \alpha_k \mathrm{sign}(x + b_k) - f(x) \right|.
\end{aligned}
\tag{46}
$$

Similarly, for the first term, we set $\alpha_k^* = \max\limits_{j \neq k} |\alpha_j|$ and $\lambda_k = \dfrac{1}{\delta_0} \ln \dfrac{(N-1)\alpha_k^* - \varepsilon_2}{\varepsilon_2}$. For $\lambda \geq \lambda_k$, we have:

$$
\frac{1}{2} \left| \sum_{j \neq k} \alpha_j s_j(x) - \sum_{j \neq k} \alpha_j \mathrm{sign}(x + b_j) \right| < \varepsilon_2.
\tag{47}
$$

For the second term, notice that when $\dfrac{k}{N} - \delta_0 \leq x \leq \dfrac{k}{N} + \delta_0$, we have:

$$
\hat{G}(x) = \begin{cases}
f\left(\dfrac{2k-1}{2N}\right), & \dfrac{k}{N} - \delta_0 \leq x < \dfrac{k}{N} \\[2mm]
\dfrac{1}{2} f\left(\dfrac{2k-1}{2N}\right) + \dfrac{1}{2} f\left(\dfrac{2k+1}{2N}\right), & x = \dfrac{k}{N} \\[2mm]
f\left(\dfrac{2k+1}{2N}\right), & \dfrac{k}{N} < x \leq \dfrac{k}{N} + \delta_0,
\end{cases}
\tag{48}
$$

and

$$
\alpha_k \mathrm{sign}(x + b_k) = \begin{cases}
\left[ f\left(\dfrac{2k-1}{2N}\right) - f\left(\dfrac{2k+1}{2N}\right) \right], & \dfrac{k}{N} - \delta_0 \leq x < \dfrac{k}{N} \\[2mm]
0, & x = \dfrac{k}{N} \\[2mm]
\left[ f\left(\dfrac{2k+1}{2N}\right) - f\left(\dfrac{2k-1}{2N}\right) \right], & \dfrac{k}{N} < x \leq \dfrac{k}{N} + \delta_0.
\end{cases}
\tag{49}
$$

We thus have

$$
\hat{G}(x) - \frac{1}{2} \alpha_k \mathrm{sign}(x + b_k) = \frac{1}{2} \left[ f\left(\frac{2k+1}{2N}\right) + f\left(\frac{2k-1}{2N}\right) \right], \quad \text{for } \frac{k}{N} - \delta_0 \leq x \leq \frac{k}{N} + \delta_0.
\tag{50}
$$

Following Eqn.34, we replace with the new $s_j(x)$. We obtain that:

$$
\begin{aligned}
&\left| \hat{G}(x) - \frac{1}{2}\alpha_k \text{sign}(x + b_k) + \frac{1}{2}\alpha_k s_k(x) - f(x) \right| \\
&= \left| \frac{1 + s_k(x)}{2} f\left( \frac{2k+1}{2N} \right) + \frac{1 - s_k(x)}{2} f\left( \frac{2k-1}{2N} \right) - f(x) \right| \\
&\leq \left| \frac{1 + s_k(x)}{2} \right| \left| f\left( \frac{2k+1}{2N} \right) - f(x) \right| + \left| \frac{1 - s_k(x)}{2} \right| \left| f\left( \frac{2k-1}{2N} \right) - f(x) \right| \\
&\leq \frac{1 + s_k(x)}{2} \cdot L \cdot \left( \frac{2k+1}{2N} - x \right) + \frac{1 - s_k(x)}{2} \cdot L \cdot \left( x - \frac{2k-1}{2N} \right) \\
&= \frac{1}{2} L \cdot \frac{1}{N} + \frac{s_k(x)}{2} \cdot L \cdot \left( \frac{2k+1}{2N} - x \right) + \frac{s_k(x)}{2} \cdot L \cdot \left( \frac{2k-1}{2N} - x \right) \\
&= \frac{L}{2N} + L s_k(x) \left( \frac{k}{N} - x \right) \\
&\leq \frac{L}{2N} \\
&\leq \varepsilon - \varepsilon_2,
\end{aligned}
\tag{51}
$$

for the new $s_k(x)$ also satisfies that $s_k(x) \in (-1, 1)$, and $s_k(x)\left( \dfrac{k}{N} - x \right) \leq 0$.

Then we get $|G(x) - f(x)| < \varepsilon, \forall x \in [0, 1]$.

Therefore, given $\lambda^* = \max(\lambda_1, \lambda_2, \cdots, \lambda_N)$, when $\lambda \geq \lambda^*$, we have $|G(x) - f(x)| < \varepsilon, \forall x \in [0, 1]$. $\qquad \square$

### A.4.2. TANH

Since $\tanh(x) = 2\sigma(2x) - 1$, the proof is almost the same as that of sigmoid.

## A.5. Proof of Proposition 3

**Proposition 3.** (Approximation Bound of ReLU) *Given $\mathcal{F} = \mathcal{F}([0,1]; L)$ and $\mathcal{G} = \mathcal{G}(N; ReLU)$, where $ReLU(x) = \max(0, x)$ denotes the ReLU function. Given the error bound $\varepsilon > 0$, we have*

$$
\lfloor L/2\varepsilon \rfloor - 1 \leq \mathcal{N}(ReLU) \leq \lfloor L/2\varepsilon \rfloor + 2.
\tag{52}
$$

*Similarly, we have $\lfloor L/2\varepsilon \rfloor - 1 \leq \mathcal{W}(ReLU) \leq \lfloor L/2\varepsilon \rfloor + 2$.*

### A.5.1. UPPER BOUND

*Proof.* Given $G(x) = \sum\limits_{j=1}^{N} \alpha_j ReLU(w_j x + b_j) \in \mathcal{G}(N; ReLU)$, where $N = \lfloor L/2\varepsilon \rfloor + 2$. To begin with, we give the target function of $G(x)$ as $\hat{G}(x)$. Here we denote $\hat{G}(x)$ as

$$
\hat{G}(x) = N \left[ f\left( \frac{j}{N} \right) - f\left( \frac{j-1}{N} \right) \right] \left( x - \frac{j-1}{N} \right) + f\left( \frac{j-1}{N} \right), \text{ for } \frac{j-1}{N} \leq x < \frac{j}{N},
\tag{53}
$$

where $j$ satisfies that $1 \leq j \leq N$. Meanwhile, we let $\hat{G}(1) = f(1)$.

Here we prove that $|\hat{G}(x) - f(x)| < \varepsilon$ for $x \in [0, 1]$. For $\dfrac{j-1}{N} \leq x < \dfrac{j}{N}$ where $1 \leq j \leq N$, we have:

$$
\begin{aligned}
|f(x) - \hat{G}(x)| &= \left| f(x) - N \left[ f\left(\frac{j}{N}\right) - f\left(\frac{j-1}{N}\right) \right] \left(x - \frac{j-1}{N}\right) - f\left(\frac{j-1}{N}\right) \right| \\
&= \left| f(x) - f\left(\frac{j-1}{N}\right) - N\left[f(x) - f\left(\frac{j-1}{N}\right)\right]\left(x - \frac{j-1}{N}\right) - N\left[f\left(\frac{j}{N}\right) - f(x)\right]\left(x - \frac{j-1}{N}\right) \right| \\
&= \left| N\left[f(x) - f\left(\frac{j-1}{N}\right)\right]\left(\frac{j}{N} - x\right) + N\left[f(x) - f\left(\frac{j}{N}\right)\right]\left(x - \frac{j-1}{N}\right) \right| \\
&\leq N\left|f(x) - f\left(\frac{j-1}{N}\right)\right|\left(\frac{j}{N} - x\right) + N\left|f(x) - f\left(\frac{j}{N}\right)\right|\left(x - \frac{j-1}{N}\right).
\end{aligned}
\tag{54}
$$

Both $f(x)$ and $\hat{G}(x)$ are $L$-Lipchitz continuous functions. Therefore, we obtain that:

$$
\begin{aligned}
|f(x) - \hat{G}(x)| &\leq N \cdot L\left(x - \frac{j-1}{N}\right) \cdot \left(\frac{j}{N} - x\right) + N \cdot L\left(\frac{j}{N} - x\right) \cdot \left(x - \frac{j-1}{N}\right) \\
&\leq 2NL\left(x - \frac{j-1}{N}\right)\left(\frac{j}{N} - x\right) \\
&\leq 2NL \cdot \frac{1}{4N^2} \\
&< \varepsilon.
\end{aligned}
\tag{55}
$$

Now we prove that, there is a $G(x) \in \mathcal{G}(\lfloor L/2\varepsilon \rfloor + 2; ReLU)$, such that $G(x) = \hat{G}(x)$. For $G(x) = \sum_{j=1}^{N} \alpha_j ReLU(w_j x + b_j)$, we set

$$
\begin{cases}
\alpha_1 &= Nf(0) \\
\alpha_j &= N\left[ f\left(\frac{j-1}{N}\right) - f\left(\frac{j-2}{N}\right) \right] - \sum_{i=1}^{j-1} \alpha_i, \text{ for } 2 \leq j \leq N \\
w_j &= 1, \text{ for } 1 \leq j \leq N \\
b_j &= -\frac{j-2}{N}, \text{ for } 1 \leq j \leq N.
\end{cases}
\tag{56}
$$

Therefore, we have $G(x) = \hat{G}(x)$. Then we further get that $|G(x) - f(x)| < \varepsilon$ for $x \in [0, 1]$. $\qquad\square$

### A.5.2. LOWER BOUND

*Proof.* To prove that $\lfloor L/2\varepsilon \rfloor - 1 \leq \mathcal{N}(ReLU)$, we can just prove that: there is a $f(x) \in \mathcal{F}([0, 1]; L)$, that can not ensure that $|G(x) - f(x)| < \varepsilon$ in $[0, 1]$ for all $G(x) \in \mathcal{G}(\lfloor L/2\varepsilon \rfloor - 2; ReLU)$.

We construct $f(x)$ as follows:

$$
f(x) = \begin{cases}
-\varepsilon + 2\varepsilon Nx, & \text{if } 0 \leq x < \dfrac{1}{N}, \\
3\varepsilon - 2\varepsilon Nx, & \text{if } \dfrac{1}{N} \leq x < \dfrac{2}{N}, \\
f\left(x - \dfrac{2}{N}\right), & \text{if } \dfrac{2}{N} \leq x \leq 1.
\end{cases}
\tag{57}
$$

Here $N = \lfloor L/2\varepsilon \rfloor$. For $N \leq L/2\varepsilon$, it is easy to identify that $f(x)$ is $L$-Lipchitz continuous in $[0, 1]$.

Now, we assume that there is a $G(x) \in \mathcal{G}(\lfloor L/2\varepsilon \rfloor - 2; ReLU)$, subjected to $|f(x) - G(x)| < \varepsilon$ for $x \in [0, 1]$. Then for $x = 0, \dfrac{1}{N}, \dfrac{2}{N}, \cdots, 1$, they all satisfy that $|f(x) - G(x)| < \varepsilon$. Specially, we obtain that $f(0) = f(2/N) = f(4/N) = \cdots = -\varepsilon$, and $f(1/N) = f(3/N) = \cdots = \varepsilon$. We further obtain that $G(0), G(2/N), G(4/N), \cdots$ are all negative, while $G(1/N), G(3/N), \cdots$ are all positive. Conclusively, we have $(-1)^k G(k/N) < 0$.

Here we analysis $G(x)$ on each interval $\left(\dfrac{k-1}{N}, \dfrac{k}{N}\right)$ with Lagrange's Mean Value Theorem.

Since $G(x) = \displaystyle\sum_{j=1}^{N-2} \alpha_j ReLU(w_j x + b_j)$, we know $G(x)$ is differentiable except $x = -b_j/w_j (w_j \neq 0)$ where $j = 1, 2, \cdots, N-2$.

Here we prove that: there is some $x_k \in \left(\dfrac{k-1}{N}, \dfrac{k}{N}\right)$, such that $(-1)^k G'(x_k) < 0$.

1) If $G(x)$ is differentiable in $\left(\dfrac{k-1}{N}, \dfrac{k}{N}\right)$, for $(-1)^k G\left(\dfrac{k-1}{N}\right) > 0, (-1)^k G\left(\dfrac{k}{N}\right) < 0$, we obtain

$$G'(x_k) = \left[G\left(\dfrac{k}{N}\right) - G\left(\dfrac{k-1}{N}\right)\right]/(1/N). \tag{58}$$

for some $x_k \in \left(\dfrac{k-1}{N}, \dfrac{k}{N}\right)$, by Lagrange's Mean Value Theorem.

Furthermore, we have:
$$\begin{aligned}
(-1)^k G'(x_k) &= (-1)^k N \left[G\left(\dfrac{k}{N}\right) - G\left(\dfrac{k-1}{N}\right)\right] \\
&= N\left[(-1)^k G\left(\dfrac{k}{N}\right) - (-1)^k G\left(\dfrac{k-1}{N}\right)\right] \\
&< 0.
\end{aligned} \tag{59}$$

2) If $G(x)$ is not differentiable, there must be $\Theta \subseteq \{-b_j/w_j : w_j \neq 0, j = 1, 2, \cdots, N-2\}$, such that $\Theta \in \left(\dfrac{k-1}{N}, \dfrac{k}{N}\right)$.

At least, we know $G(x)$ is continuous in $\left(\dfrac{k-1}{N}, \dfrac{k}{N}\right)$, and not differentiable only in $\Theta$. We assume $\Theta = \{b_1', b_2', \cdots, b_m'\}$ (they are different from each other), and $\dfrac{k-1}{N} < b_1' < b_2' < \cdots < b_m' < \dfrac{k}{N}$. Then one of the following formulas holds—

$$\begin{aligned}
(-1)^k G\left(\dfrac{k-1}{N}\right) - (-1)^k G(b_1') &> 0, \\
(-1)^k G(b_1') - (-1)^k G(b_2') &> 0, \\
\cdots, & \\
(-1)^k G(b_{m-1}') - (-1)^k G(b_m') &> 0, \\
(-1)^k G(b_m') - (-1)^k G\left(\dfrac{k}{N}\right) &> 0
\end{aligned} \tag{60}$$

—otherwise we will obtain that $(-1)^k G\left(\dfrac{k-1}{N}\right) \leq (-1)^k G\left(\dfrac{k}{N}\right)$, which contradicts the assumption.

Therefore, we can apply Lagrange's Mean Value Theorem with the established formula above, we can find some $x_k \in \left(\dfrac{k-1}{N}, \dfrac{k}{N}\right)$, such that $(-1)^k G'(x_k) < 0$.

Next, we show one important property of $G(x)$. If $a < b$ and $G'(a) \neq G'(b)$, we will find some $j$, such that $a < -b_j/w_j < b$. This is because that $G''(x) = 0$ holds almost everywhere, except $x \in \{-b_j/w_j : w_j \neq 0, j = 1, 2, \cdots, N\}$—if there is no $j$ satisfying that $a < -b_j/w_j < b$, we will have $G'(a) = G'(b)$.

Furthermore, since we have obtained that $G'(x_1) > 0, G'(x_2) < 0, G'(x_3) > 0, \cdots$, for each interval $(x_1, x_2), (x_2, x_3), \cdots, (x_{N-1}, x_N)$, there must be some $N-1$ different $j_k$, such that $-b_{j_k}/w_{j_k} \in (x_k, x_{k+1})$ for $k = 1, 2, \cdots, N-1$. However, only $N-2$ different $j$ are available in $G(x) = \displaystyle\sum_{j=1}^{N-2} \alpha_j ReLU(w_j x + b_j)$. By the

pigeonhole principle, we can not find a $G(x) \in \mathcal{G}(\lfloor L/2\varepsilon \rfloor - 2; ReLU)$, subjected to $|f(x) - G(x)| < \varepsilon$ for $x \in [0, 1]$.

Therefore, we get the minimum $\mathcal{N}(ReLU) \geq \lfloor L/2\varepsilon \rfloor - 1$. $\qquad\square$

## B. Experiments

### B.1. Approximation Experiments

#### B.1.1. APPROXIMATION LANDSCAPES

In this section, we will conduct a detailed analysis of the experimental results mentioned in Section 4.2. The experimental setup remains consistent with that described in Section 4.2.

Tables I and II present how the fitting performance of the PLN and PLS activation functions varies with changes in network width under different norm sizes. From these tables, it can be observed that as the network width increases, the model's approximation performance gradually improves. These results indicate that increasing the network width can enhance the model's approximation capabilities.

Table III further compares the two best-performing activation functions, PLN-4 and PLS-2, with other activation functions. It can be seen that sigmoid and tanh perform better in approximating the target function, while ReLU's performance is relatively poor. PLN-4 also exhibits good approximation ability, especially at smaller network widths. These results suggest that different activation functions have varying approximation performances at different network widths.

*Table I.* Approximation landscapes with PLN.

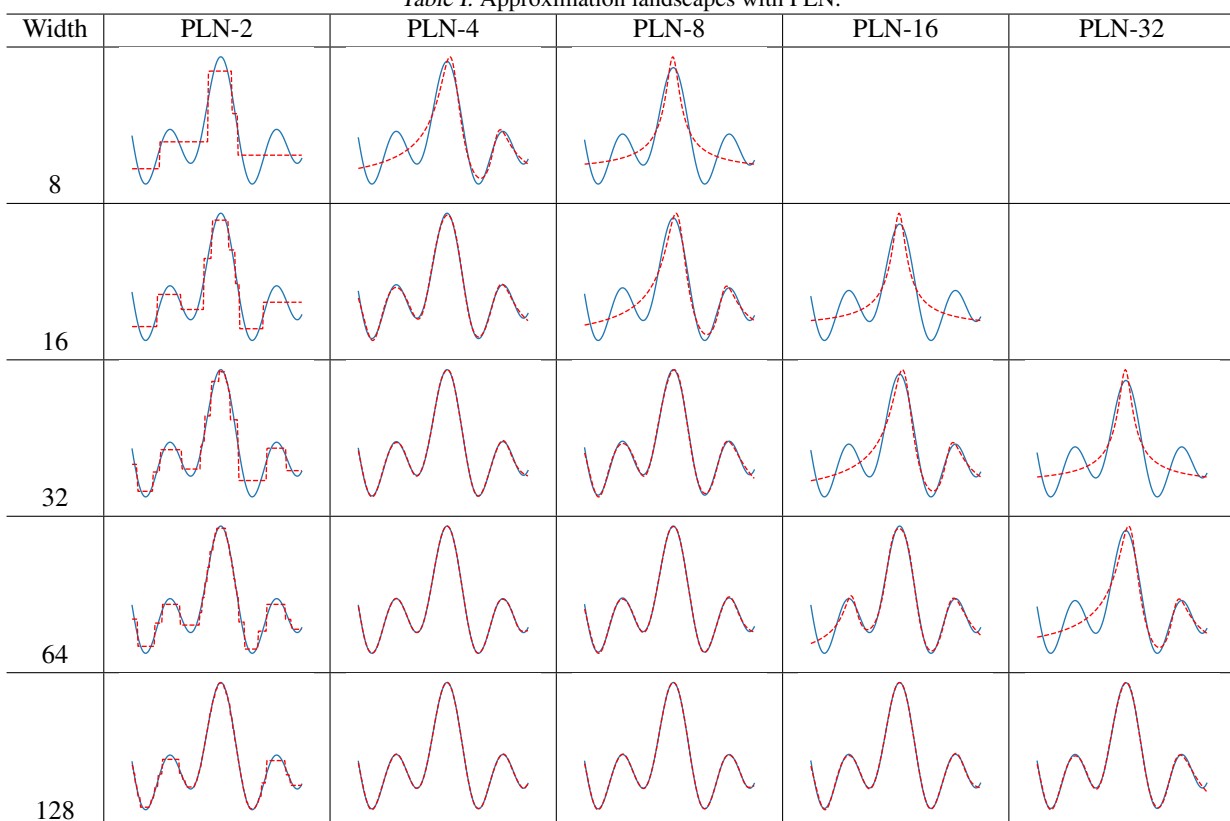

| Width | PLN-2 | PLN-4 | PLN-8 | PLN-16 | PLN-32 |
|-------|-------|-------|-------|--------|--------|
| 8 | | | | | |
| 16 | | | | | |
| 32 | | | | | |
| 64 | | | | | |
| 128 | | | | | |

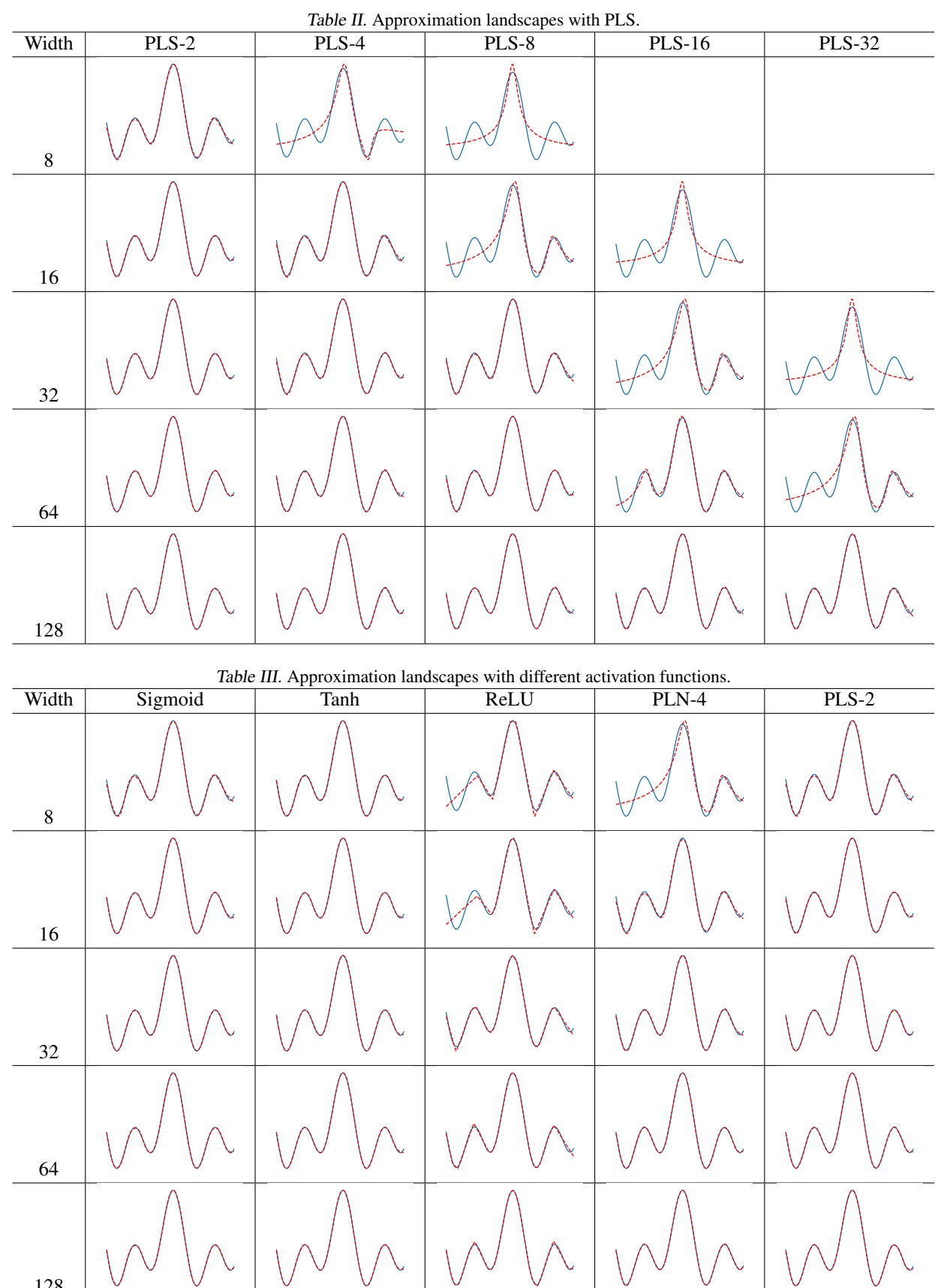

*Table II.* Approximation landscapes with PLS.

*Table III.* Approximation landscapes with different activation functions.

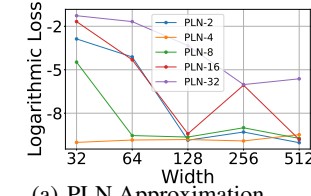
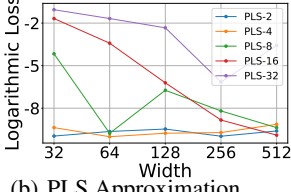

(a) PLN Approximation.     (b) PLS Approximation.

*Figure A1.* The results of logarithmic loss of PLN and PLS on random functions varying with width.

### B.1.2. APPROXIMATION RESULTS

In the main text Figure 3, we presented the best logarithmic loss of PLN and PLS with different norm sizes when fitting trigonometric functions. Here, we provide the fitting results in Figure A1 for random functions as a supplement, and the target function is

$$y(x) = 2 \cdot \text{rand}(x) - 1, \quad x \in \{-5, -4.5, \ldots, 5\} \quad (61)$$

where $\text{rand}(\cdot)$ follows a uniform distribution in the range $[0, 1]$.

Analyzing the figures readily reveals that in the fitting task of random functions, PLN-4 and PLS-2 perform the best, which is consistent with the results presented in the main text.

### B.2. Classification with CNN

#### B.2.1. EXPERIMENT SETTINGS

**Experiment Settings.** We conduct the classification experiments on CIFAR-10 dataset using VGG-16. We set the width (or the channel number) of each hidden layer to be the same for simplification. Here we set the width as 64. We vary the activation functions in sigmoid, tanh, ReLU, PLN and PLS. We train a total of 240 epochs using SGD with a mini-batch size of 128, momentum of 0.9 and weight decay of 0.0001. The initial learning rate is set to 0.1, and divided by 2.5 at the 60th, 100th, 140th, 180th and 220th epochs. We use warmup in the first 20 epochs. We also use data augmentation. We record the average results among three random seeds.

#### B.2.2. EXPERIMENTS ON NORM SIZE

**Experiments on norm size.** Norm size ($d$) is a hyperparameter in PLN-$d$ and PLS-$d$. We fix the width as 128, $d$ ranges in $2, 4, 8, 16, 32, 64, 128$. The results are shown in Figure A2.

Figures A2 and A3 in the appendix detail the performance of PLN and PLS activation functions with varying norm sizes on the CIFAR-10 dataset using two different CNN architectures: VGG-16 and ResNet20. Both figures are split into training and test accuracy plots, with the x-axis

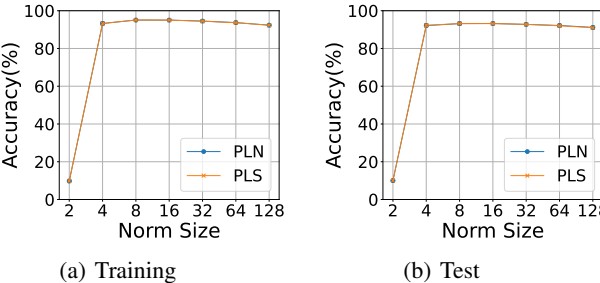

(a) Training        (b) Test

*Figure A2.* Results of PLN and PLS with width 128 and different norm sizes on CIFAR-10 using VGG-16 with BN.

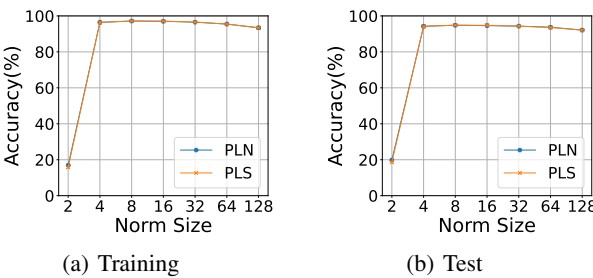

(a) Training        (b) Test

*Figure A3.* Results of PLN and PLS with width 128 and different norm sizes on CIFAR-10 using ResNet-20.

representing the norm size and the y-axis showing accuracy percentage. For both architectures, PLN and PLS show a sharp increase in accuracy as the norm size increases from 2 to 4, after which accuracy plateaus. This indicates that a norm size of 4 or greater is sufficient for optimal performance, and increasing the norm size further does not significantly improve accuracy.

The results demonstrate that PLN and PLS perform similarly across different norm sizes, achieving high accuracy with both VGG-16 and ResNet20 on CIFAR-10. These findings suggest that larger norm sizes are not necessary for achieving good performance with these activation functions in CNNs.

#### B.2.3. EXPERIMENTS ON WIDTH

In this section, we supplement the experimental results using the ResNet-20 network architecture on the CIFAR-10 dataset to further verify the performance of different activation functions across varying network widths. Figure A5 illustrates how the accuracy of ReLU, PLN-8, Sigmoid, Tanh, and PLS-8 activation functions changes with network width on both the training and test sets. The ReLU activation function demonstrates higher accuracy across all widths, with PLN-8 and PLS-8 showing comparable performance. The Sigmoid and Tanh activation functions exhibit relatively lower performance. The overall performance ranking is ReLU > PLN-8 $\approx$ PLS-8 > Tanh > Sigmoid, which is consistent with the results presented in the main text.

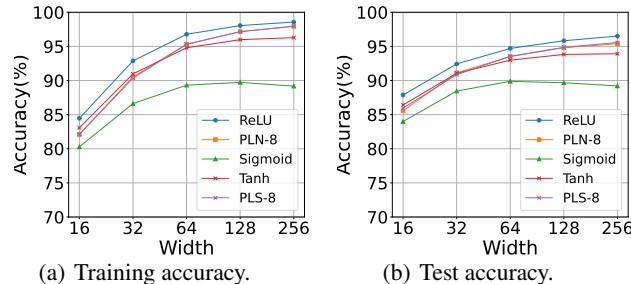

(a) Training accuracy.         (b) Test accuracy.

*Figure A4.* Results of different activation functions with different widths on CIFAR-10 using resnet20.

### B.3. Experiments on Transformer

#### B.3.1. MACHINE TRANSLATION WITH TRANSFORMER

In the translation task training process, each task is trained for 100 epochs, with the first 10 epochs utilizing a warmup strategy and the remaining 90 epochs following a cosine decay learning rate schedule. The maximum learning rate is set to 5e-4, and the optimizer used is Adam with a weight decay of 5e-4. Each task is conducted using three different random seeds (10, 20, and 30), and the final results are averaged. All experiments are conducted on an NVIDIA RTX 3090 GPU, with each task taking approximately 50 minutes to complete.

#### B.3.2. LONG TIME SERIES PREDICTION TASKS WITH TRANSFORMER

All the experiments are implemented in PyTorch (Paszke et al., 2019) and conducted on a single NVIDIA A40 40GB GPU. We utilize ADAM (Kingma & Ba, 2014) with an initial learning rate at $5 \times 10^{-4}$ and L2 loss for the model optimization. The batch size is uniformly set to 32 and the number of training epochs is fixed to 10. We set the number of Transformer encoder layers to 3 and decoder layers to 2. In order to more accurately determine the impact of the normalization layer and activation layer on the network, we used the Traffic dataset with a data dimension of 862 and a total length of 17544 for the experiment. We extended the sequence length that the model needs to process at a time from 96 to 720, and the prediction sequence length is still set to 720.

