# OpenReview forum: "Universal Approximation Theorem of Networks Activated by Normalization"
_ICML.cc/2025/Conference — Submitted to ICML 2025_

### Official Review · Reviewer_G246 · 2025-02-23

**Overall Recommendation:** 4

**Summary:**

The authors study the approximation power of MLPs with no traditional activation functions but instead only with the *layer norm* between affine layers.  They show that this too is a universal approximator.   Personally, I find this result interesting as it is something I've wondered about showing my self but never got around to investigating it; nice job! :)

Their analysis is rigorous, cleanly explained, and accompanied but some numerical illustrations (though the numerics are largely vaguely supported but welcome).

---
That said, I have some questions/concerns-ish (below).

**Claims And Evidence:**

Rigorous proofs.

**Essential References Not Discussed:**

The authors have no connection to classical non-linear approximation theory; specifically non-linear (manifold) widths; e.g. [1].  Recent results such as [2] would be relevan.


[1] DeVore, Ronald A. "Nonlinear approximation." Acta numerica 7 (1998): 51-150.

[2] Cohen, A., and R. DeVore. "Nonlinear Approximation and (Deep) ReLU Networks." Constructive Approximation (2021).

**Experimental Designs Or Analyses:**

Largely irrelevant.

**Methods And Evaluation Criteria:**

NA

**Other Comments Or Suggestions:**

Definition 3 - In the *approximation theory* literature, which you are aiming for, this is a *width* of $\mathcal{F}$.  For instance, if $\mathcal{G}$ were the set of all $N$-dimensional linear subspaces of a Banach space containing $\mathcal{F}$ then this quantity is the Kolmogorov linear width.

**Other Strengths And Weaknesses:**

Quantitative estimates would have been nice, but that's for a figure time and place perhaps...

**Questions For Authors:**

1. Can the authors clarify what is rigersouly meant by "optimization capacity" in this sentence: "LN-2 may have more representation capacity than PLN-4 in theory, but the optimization capacity is less?" (Also bad grammar).


2. I don't really see the point of section 4 personally. The authors consider the *width* of the class of L-Lipschitz functions on the unit interval, there is no effect of dimensional explored so I don't really get the point (clearly any such function can very easily be approximated by a piecewise linear interpolation with no piece having slope more than L); what am I missing?

3. Together Proposition 1 and 3 don't really conclude anything.  There is an upper bound for NNs with LN which does not beat the lower bound for NNs with ReLU non-linearity.  So there is some evidence but no mathematical conclusion..  Why include this as it is a proof of little....

**Relation To Broader Scientific Literature:**

Very nice!

**Theoretical Claims:**

Correct, and rigorously proven.

---

> ### Author Rebuttal · Authors · 2025-04-01
>
> ### Reply to the reviewer G246
>
> Thanks for your valuable comments and suggestions. We are pleased with your support for our paper.
>
> ---
>
> #### **Response to Question 1**
>
> Thanks for figuring out this typo. We indeed want to express "better optimization property" with the words "optimization capacity". A better optimization property of a module means that, adding the module to a network may improve conditioning, numerical stability and training efficiency in the optimization process. The optimization property of normalization is the main reason why normalization can be widely applied in various deep neural networks, as we described in introduction of this paper.
>
> Here we take PLN-2 as an example. When LN acts on $R^2$, LN constrains the mean and variance both and it outputs only two values. This constrain is compact for only two neurons, ensuring that the output is nearly constant---assuming the input vector is $(x_1,x_2)$ and $x_1>x_2$, we can identify that the output must be a constant vector $(1,-1)$ as long as $x_1>x_2$. Consequently, the derivative of the output with respect to $x_1$ is zero, which can lead to stagnation in parameter updates during optimization. This observation explains why PLN-2 has " worse optimization property", despite our theoretical proof of its strong approximation capabilities. Therefore, we believe that there is a trade-off between optimization and optimization, as shown in Section 4.
>
> ---
>
> #### **Response to Question 2**
>
> In Section 4, we aim to provide a comprehensive understanding of the theoretical and practical aspects of activation functions. Specifically:
>
> 1. **Theoretical Comparison**: In Section 4.1, we analyze the approximation bounds of different activation functions, including PLN and ReLU. Our theory aims to provide quantitative results of different activation functions, which can be regarded as a theoretical reference when we explore further in experiment.
>
> 2. **Practical Insights**: In practical scenarios, the parameters of the network is obtained by training, resulting the network may perform much worse than its theoretical case. Figure 5(c) is a supportive example for this. Therefore, it is necessary to consider the optimization property of activation functions.
>
> Our ultimate goal is to demonstrate that a network with PLN is **not only a universal approximator** but **also inherits the optimization benefits** of normalization. This dual property of PLN—combining strong approximation capabilities with improved optimization dynamics—makes it a promising candidate for simplifying deep neural network architectures and advancing their theoretical understanding.
>
> We also add approximation experiments with 512 random inputs and labels on $R^8\times R$ to explore the high-dimensional case, following the same experimental settings in section 4.1. Here we give the MSE loss using a network with depth 1 and width 256.
>
> | PLN-4    | PLS-4    | Sigmoid | Tanh    | ReLU    |
> | -------- | -------- | ------- | ------- | ------- |
> | 3.78e-12 | 2.01e-10 | 2.14e-3 | 5.17e-8 | 2.97e-6 |
>
> We find PLN-4 and PLS-4 performs much better than the other activation functions, beyond the results of $R \to R$ approximation experiments in this paper.
>
> Additionally, we have included experiments on high-dimensional data in Section 5. While these experiments are not focused on approximation tasks, they further highlight the practical advantages of PLN in deep learning contexts.
>
> ---
>
> #### **Response to Question 3**
>
> In Section 4.1, our theoretical analysis provides quantitative insights into the approximation capabilities of different activation functions. However, as demonstrated in Section 4.2, the practical performance of a network is also heavily influenced by its optimization properties.
> As you can see, our experiment is conducted on a simple network. Although we applied various training techniques (line 252-262) to attempt to find the better parameters, it is still a hard task---referring to the bad performance in Figure 5(c) . We find the optimization problem exists, even in such a tiny network (depth 1, width 16). Therefore, We further explore the relationship between optimization and approximation in deep neural networks, namely the experiments in section 5.
>
> The propositions in Section 4.1 give us a good reference of the approximation capacity of a network. Only based on Section 4.1, we can contribute some bad performances of a network to the optimization problem rather than its approximation capacity **confidently**---based on that we know the theoretical performance of this network is outstanding, but we have not exploit it in a certain training process.
>
> ---
>
> #### **About Essential References**
>
> Upon carefully reviewing, we find the two references provide are supportive work for this paper. These will be added to citations in revision. We sincerely appreciate the reviewer for mentioning them.
>
> ---
>
> Much thanks for your support again.

---

### Official Review · Reviewer_1Q2E · 2025-03-10

**Overall Recommendation:** 1

**Summary:**

The authors study universal approximation for networks, where the activation function is replaced by a layer normalization. This result is traced back to the classical universal approximation theorem by Cybenko. However, this step contains an error, as the sigmoid function derived from  LN does not act element wise on the output of the affine transformation of the layer. Therefore the utilization of Cybenko's result is not legitimate as the pre-conditions are not met. I therefore find that at the present stage of preparation, the paper is not yet suitable for publication.
Also, the 1d input dimension results do not really change this finding, as their (practical) scope is very limited.
This said, I do not intend to claim that the statement of the theorem the authors give is wrong. I can well imagine that it is correct as a mathematical statement, as there are many activation functions known that do not act element wise. Nevertheless, it is not yet proven in the present version of the article.
This finding can not be compensated by the numerical experiments the authors provide. Though the 1-d experiments are convincing,  the further experiments on the VGG architecture and CIFAR10 and the time series task are much less. In particular, the results the authors have obtained up to now, do not really give a sound experimental basis to support the author's claims.

# # Thanks for the discussion , but my opinion stays unchanged, as the way the authors define their normalization layers does not really match with what generally is understood as layer normalization. The paper seems formally correct, but the result is too limited.

**Claims And Evidence:**

As indicated above, the application of Theorem 1 in the proof of Theorem 2 contains an error, as \sigma(x) in the last line of the proof does not act element wise, as required in the original version of Theorem 1. Therefore, the proof f Theorem 2 requires adjustment. This might be feasible, but as this is somewhat the core of the paper, this should lead to a rejection for this time.

**Essential References Not Discussed:**

None

**Experimental Designs Or Analyses:**

Weak, see above under Methods and Evaluation Criteria.

**Methods And Evaluation Criteria:**

The experiments on the 1d example and CIFAR10 and the time series task are inconclusive and are not yet suited to promote the proposed architecture.

**Other Comments Or Suggestions:**

The subject per se is not uninteresting. i encourage the authors to correct their proof and improve the paper throughout, including creating a reasonaby solid experimental basis, and then resubmit.

**Other Strengths And Weaknesses:**

The status of preparation of the paper is very preliminary and there are too many typos and small errors to list them here. The authors should conduct a careful reediting of the paper, before submitting it elsewhere.

**Questions For Authors:**

None

**Relation To Broader Scientific Literature:**

I'm not an expert on UAP with 'exotic' activations.

**Theoretical Claims:**

Checked proof of main Thm 2, which contains an error.

---

> ### Author Rebuttal · Authors · 2025-04-01
>
> ### Reply to the reviewer 1Q2E
>
> Thanks for your valuable comments and suggestions.
>
> The main concern raised by the reviewers is the correctness of our proof. After re-examining our proof, we are confident that there are **no errors** in our reasoning. Here, we attempt to clarify why the reviewer might have misunderstood our proof.
>
> ---
>
> #### **Clarification on the Misunderstanding of the Proof**
>
> The reviewer pointed out that the operation $\sigma$ does not act element-wise. We trace back to lines 132–155 and clarify the following:
>
> In line 150, $\sigma(\boldsymbol{w^\top_j x} + b_j)$ acts element-wise. Here, both $\boldsymbol{w}$ and $\boldsymbol{x}$ are vectors, and $b_j$ is a scalar. Therefore, $\boldsymbol{w^\top_j x} + b_j$ is a real number rather than a vector. Consequently, $\sigma(\boldsymbol{w^\top_j x} + b_j)$ is applied element-wise. As we mentioned in line 153, $\sigma(x) = (x / |x| + 1) / 2$ is a function on $R$. Besides, the form in our proof (line 150) aligns with Cybenko's work (line 65, Eqn. 1), ensuring the correctness of our proof.
>
> If this is not the source of confusion, another possible reason is the discrepancy between the input and output dimensions of the Layer Normalization (LN) operation. In our proof, we construct $G(x) = \sum\limits_{j=1}^{N+1} \boldsymbol{\alpha}_j^\top LN(\boldsymbol{W_j x} + b_j)$, where $\boldsymbol{W}_j$ and $\boldsymbol{b}_j$ belong to the first linear layer, and $\boldsymbol{\alpha}_j$ belongs to the second linear layer. We set $d = 2$ in this proof, as mentioned in the paper.
>
> The input to the $N+1$ LNs has $2(N+1)$ neurons, and the output of the LNs also has $2(N+1)$ neurons. However, the final output of Eqn.5 is the linear combination of $N$ neurons. This is because:
>
> 1. We merge the $(N+1)$-th term into the previous $N$ terms.
> 2. We set $\boldsymbol{\alpha}_j = [\hat{\alpha}_j, 0]^\top$, as shown in line 136. While the previous $N$ terms output $2N$ neurons, only $N$ of them are allowed to pass—because the other $N$ neurons are multiplied by zero in $\boldsymbol{\alpha}_j$. This is why the reviewers might have misunderstood our proof.
>
> If the reviewer still has trouble understanding our proof. Please feel free to point it out, and we would like to clarify it.
>
> ---
>
> #### **Clarification on the Experimental Design**
>
> The experiments are not solely designed to support the theory, they also lead to a discussion on approximation and optimization. In practical scenarios, the parameters of the network is obtained by training, resulting the network may perform much worse than its theoretical case. Figure 5(c) is a supportive example for this. Therefore, it is necessary to consider the optimization property of activation functions. In this paper, we find that PLN can be seen as a combination of normalization and activation—possessing both good approximation properties (as we propose) and optimization properties (inherited from normalization). As we concluded in Section 5, deep neural networks with only linear modules and PLN can perform well. These experiments identified both the approximation and optimization properties of PLN.

---

> > ### Comment · Reviewer_1Q2E · 2025-04-05
> >
> > Tank you for this explanation. I acknowledge that in Lemma line 150 you reach a shape that is consistent with applying Cybenko's theorem. But I still don't understand how you are getting there. You define LN(x) in equation (3), which intermingles all x_j in the denominator. You then apply Layer-Normalization on sub-streams, which is made possible by the convention (4), which is not what people would normally understand as layer normalization, as this 'normalizes' just two values produced in a very spacial way with the w_j and-w_j in the matrices. From that you obtain the element wise non-polynomial activation required by Pinkus theorem.
> >
> > But this is not Layer-Normalization. Layer normalization would be to apply LN to Wx+b.
> >
> > Therefore my misunderstanding was somewhat induced by the notions you chose and the title of your paper. And I find the result, which I now understand better, much more limited because the freedom that is utilized for the 'tiny sub-layer normalization' is so wide, that not much can be learned from your observation. So I'm not yet ready to adjust the score.

---

> > > ### Author Response · Authors · 2025-04-07
> > >
> > > We thank the reviewer for the timely feedback on our rebuttal.  We provide the further clarification for your concerns.
> > >
> > > ---
> > >
> > > #### **Clarification for the misunderstanding of proof.**
> > >
> > > Thanks for your reply on our rebuttal in detail. We can infer that your misunderstanding on the proof is mainly derived from that you thought the proof is based on only one Layer Normalization (LN, formulated by Eqn. 3), while our proof is based on multiple Layer Normalizations in a layer (i.e., Parallel Layer Normalization (PLN)).  We denote PLN for simplifying the description.  We spend many words (pages) to describe and highlight the PLN. For examples, in our introduction, `Lines 061-066`, we described that---**We focus on parallel layer normalizations (PLN) rather than serial LN-Net, as shown in Figure 1. We theoretically prove an infinitely wide network—with a "linear-PLN-linear" structure—has universal approximation ability on $[0, 1]^n$**. Besides, we also introduced what is PLN, in `Lines 085-093 (Right)`, just **after the description of Layer Normalization (Eqn.3)** and **before Section 3  (obviously before Theorem 2)**.
> > >
> > > Note that PLN is a more general formulation of Group Normalization (GN) [1], which is widely used in CNNs for object detection and segmentation.  GN is also described as a more general formulation of LN in paper [1]. Besides, [2] further extended GN in CNN to LN-G in MLP, and also figured out that LN-G has more nonlinearity than LN. [2] also provided experimental basis showing that LN-G obtains better performance than LN. We highlight that  PLN-d has the same structure as LN-G in `Lines 144-155(Right)`  (please refer to Section 3.2).
> > >
> > > As for "normalizes just two values produced in a special way with the $w_j$ and $-w_j$",  note that we show that "**we give the proof in the case d=2**" in Line 132.  We also provided the case "normalizes $d$ values" in **Appendix A.1**, which is also mentioned in `Line 157`.
> > >
> > > As for the misunderstanding induced by the notions you chose and the title of the paper, does the reviewer mean the title of our Theorem 2 rather than our paper? There is no obvious "LN" in the title of our paper. As for the title of Theorem 2---"LN for UAT"---here LN can be seen as a module, and can be also seen as an operation. We consider the similar title "ReLU for UAT", a "ReLU module" has many "ReLU operations". An "LN module" has only one "LN operation", but a "PLN module" has many "LN operations". LN in the title of Theorem 2 means LN operations rather than a single LN module. **Figure 2** also shows the differences of PLN and other activation functions. This may be why the reviewer misunderstood our Theorem.
> > >
> > > ---
> > >
> > > #### **Clarification for the limited learn given the same wide.**
> > >
> > > As for the concern that our PLN-Net seems too wide to train conveniently. Please refer to Section 4 and 5 for our experiments. In section 4, although ReLU has stronger nonlinearity under the same width, **ReLU is the one hard to train rather than PLN**. In section 5.1.1, we show the advantage of PLN in deep neural networks---it possesses **both good approximation property (as we propose) and optimization property (inherit from normalization)**. Therefore, the network with PLN is easy to train.
> > >
> > > ---
> > >
> > >
> > > #### **References**
> > >
> > >  [1] Wu Y, He K. Group normalization[C]//Proceedings of the European conference on computer vision (ECCV). 2018: 3-19.
> > >
> > >  [2] Ni Y, Guo Y, Jia J, et al. On the nonlinearity of layer normalization[J]. arXiv preprint arXiv:2406.01255, 2024.

---

### Official Review · Reviewer_Guee · 2025-03-12

**Overall Recommendation:** 2

**Summary:**

This paper explores the possibility of replacing activation functions with layer normalization, offering a new perspective on the foundational logic of neural networks. It provides corresponding approximation theory, width estimates, and experimental results, including a theoretical proof of the universal approximation theorem (UAT) for linear layers equipped with layer normalization. Additionally, the paper numerically demonstrates the impact of different normalization designs on network performance.

**Claims And Evidence:**

Yes.

**Essential References Not Discussed:**

Null.

**Experimental Designs Or Analyses:**

Yes.

**Methods And Evaluation Criteria:**

Yes.

**Other Comments Or Suggestions:**

1. The footnote 1 on Page 2 is incomplete.

2. The row header of Table 3 is wrong.

3. There is a typo in Sec 3: 'Finally, we further disocuss the approximation on LN without centering' -- where 'disocuss' should be 'discuss'.

**Other Strengths And Weaknesses:**

Strengthnesses:

1. The idea of replacing the activation functions with layer normalization is interesting, especially considering the inter-neuron nonlinearity. It would be even better to derive some novel activation based on the existing normalization methods.
2. The paper presents a unique perspective by raising an important and thought-provoking question. It explores, from a theoretical standpoint, whether normalization operations can replace activation functions.

Weaknesses:

1. The theory is not particularly novel. The so-called PLS or PLN can be considered as special activation functions, and the conclusion for infinite width (Thm 2) can be directly derived from Cybenko (1989). Although the author mentions in Corollary 2 that the result can be extended to PLS, I did not see any relevant description. Moreover, LS in Corollary 2 is not the same as PLS, so I don't fully understand what conclusion Corollary 2 is trying to convey.
2. The discussion in Section 4 lacks convenience for me. The theoretical results estimate the minimum width for UAT based on 1D target functions, but they fail to provide some intrinsic differences between the proposed LN and some conventional activations. The authors also conducted some experiments to testify to the approximation capacity, but the optimization issues strongly influence the performance. The adopted experiment setting is too straightforward to distinguish the approximation and optimization properties of different activations. I think this section may need major revision, especially the experience setting part.
3. The experience conducted in Section 5 needs further discussion. The current setting includes CV and NLP models with different architectures. In some settings, the proposed PLN-8 outperformed BN (or other activations), and some don't. I think it will be interesting to discuss further the source of these preferences. The authors attempt to address this issue by assuming there is a difference in the level of nonlinearity that each task requires, but the influence of model architecture or activations (when used together) cannot be disentangled. I think this topic can even be a paper itself.

**Questions For Authors:**

1. The estimation of the approximation bound in Propositions 1-3 holds for only 1D target functions and shares a similar form of the upper bound. However, in high dimensions, will the dependence on the dimension $n$ show a difference when we choose different activations?

2. The performance of PLN seems sensitive to the hyperparameter $d$. Any criterion about the choice of $d$?

3. In Figure 10, do the results mean that compared with the identity case, adding PLN-8 can hurt the performance?

4. Figure 11 clearly shows the advantages of PLN-8, but Figure 10 does not. Is this because of the different tasks or different architecture? A possible way is to consider ViT, a transformer-based CV model, to get rid of the influence of model architecture.

**Relation To Broader Scientific Literature:**

Null.

**Theoretical Claims:**

Yes.

---

> ### Author Rebuttal · Authors · 2025-04-01
>
> ### Reply to the reviewer Guee
>
> Thanks for your valuable comments and suggestions.
>
> ---
>
> #### **Response to Weakness 1**
>
> To begin with, we list our contributions below to clarify the novelty of this paper.
>
> 1. We are the first to **consider LN (and LS) as activation functions** and provide the **mathematical proof** of its universal approximation property **by constructing proper parameters** in the linear layers.
> 2. We are the first to propose **the concept of PLN and PLS**. Although the similar concepts like GN or LN-G have been proposed in the previous work, we discussed their differences with our PLN in our paper.
> 3. This paper also discussed the **universal approximation property of normalization** for the first time.
>
> Besides, we point out that the proof is **not direct** as the reviewer commented. As we can see in the proof of Lemma 1, LN is **not equivalent to** sigmoidal functions in the network. We **construct proper weights and biases** in the linear layer and then obtain a similar result to the linear combination of sigmoidal functions.
>
> We also **construct piece wise step functions** to prove the universal approximation capacity of PLN, as shown in Appendix A.3. We initially try to show our theory of PLN in this way, but this method is more complex than the current version. Considering the readability of this paper, we decided to give the proof based on Cybenko's work.
>
> As for the reviewer's concern about Corollary 2, we provided details in the supplementary material, as we mentioned in line 197. The relationship between LS and PLS is similar to LN and PLN, as shown in Figure 1. When we put LS parally, we then obtain PLS. Compared with PLN, norm size 1 is suitable for PLS, while PLN requires the norm size of at least 2. Besides, note that LS is RMSNorm, which is also widely used in LLMs (e.g., LLama, Qwen2).
>
> ---
>
> #### **Response to Weakness 2**
>
> Due to the limited words for the reply, could you please refer to the **Response to Question 2** in the **Reply to the reviewer G246**, for understanding Section 4 better?
>
> We admit it is hard to distinguish the approximation and optimization properties of different activations. Although we applied various training techniques (line 252-262) to attempt to find the better parameters, it is still a hard task---referring to the bad performance in Figure 5(c) . We find the optimization problem exists, even in such a tiny network (depth 1, width 16). Therefore, We further explore the relationship between optimization and approximation in deep neural networks, namely the experiments in section 5.
>
> We will adjust our descriptions and expressions in the revised version for better readability.
>
> ---
>
> #### **Response to Weakness 3**
>
> This paper does not aim to show that PLN can outperform other normalizations either activation functions. We aim to show that PLN can be seen as a combination of normalization and activation---it has both good approximation property (as we propose) and optimization property (inherit from normalization). Furthermore, it may help simplify the structure and of DNNs and then provide a convenient platform to study DNNs.
>
> ---
>
> #### **Response to Question 1**
>
> There are also results for high dimensional input in the form with the sign $O(\cdot)$, as shown in the references mentioned by the reviewer G246. But it is hard to give the precise bounds like that in Section 4.1. Therefore, we add experiments to explore the high-dimensional case. Could you please refer to the **Response to Question 2** in the **Reply to the reviewer G246** for the detailed results?
>
> ---
>
> #### **Response to Question 2**
>
> We further discussed PLN-2 in the **Response to Question 1** in the **Reply to the reviewer G246**, could you please turn there for our response? Here we can see that better approximation capacity may suffer from optimization problem. As for the practical scenarios, following our experiments in section 4 and 5, we recommend that $d=4$ for shallow networks, and $d=8$ or lager for deep networks.
>
> ---
>
> #### **Response to Question 3 and 4**
> In figure 10, the "Identity" term means that we use PLN-8 as normalization and Identity as activation. The "PLN-8" term does not introduce additional nonlinearity based on the "Identity" term---it essentially add PLN-8 (as activation) after another PLN-8 (as normalization). Here we provide the results on ViT for classification with PLN-8 as Normalization.
>
> | Activation | Acc(%) |
> | ---------- | ------ |
> | Identity   | 82.77  |
> | PLN-8      | 82.51  |
> | ReLU       | 87.55  |
>
> From the classification task, we find that "PLN+ReLU" is much better than "PLN+Identity", it indicates that the architecture will affect the conclusion. On the other hand, in the time-series tasks (Figure 11), we find the performance of  "Identity-Identity" is not so bad, indicating that the task will also affect the conclusion.

---

### Decision · Program_Chairs · 2025-05-01

**Decision:**

Reject

**Comment:**

The paper explores whether neural networks composed solely of linear layers and normalization operations, specifically in a configuration referred to as Parallel Layer Normalization (PLN) can serve as universal approximators. The authors present theoretical analysis and accompanying experiments to support the claim that such architectures, even without traditional activation functions, possess strong representational capacity.

The reviewers generally agreed that the topic is novel and interesting. One reviewer found the theoretical contributions rigorous and well-motivated, and appreciated the new perspective offered by the work. However, the majority of reviewers raised substantial concerns that ultimately were not fully resolved through the rebuttal process.

A primary issue was the soundness and clarity of the central theoretical proof. The reviewers expressed concern that the proof of the main theorem relies on assumptions that do not align with the conditions of Cybenko’s universal approximation theorem. In particular, the version of layer normalization employed in the proof deviates from standard formulations, applying normalization in a narrowly defined, highly controlled way that does not match common usage. Although the authors responded with a detailed breakdown of their construction, the reviewing team remained unconvinced, viewing the deviation in terminology and setup as undermining the generality and impact of the result.

Additional concerns were raised about the scope and presentation of the work. The use of the term "Layer Normalization" was seen as somewhat misleading, since the PLN structure is not equivalent to the layer normalization commonly used in deep learning practice (even though PLN is well-defined in the paper). This makes it harder to assess if the theoretical findings have a practical contribution and whether the paper's notion of normalization has practical significance comparable to the standard layer normalization.

Overall, while the question addressed by this paper is interesting and the theoretical line of inquiry shows promise, the current submission appears to require substantial changes and clarifications before it is acceptable for publication. We encourage the authors to revise the paper by clarifying the definitions and assumptions used in the proofs, aligning the terminology with community standards, and strengthening the empirical section to better illustrate when and why PLN is advantageous in practice.